# Global atmospheric hydrogen chemistry and source-sink budget equilibrium simulation with the EMAC v2.55 model

Nic Surawski[1,2], Benedikt Steil[2], Christoph Brühl[2], Sergey Gromov[2], Klaus Klingmüller[2], Anna Martin[2], Andrea Pozzer[2,3], and Jos Lelieveld[2,3]

[1]Centre for Green Technology, University of Technology Sydney, Gadigal Country, Ultimo NSW 2007, Australia
[2]Atmospheric Chemistry Department, Max Planck Institute for Chemistry, 55128 Mainz, Germany
[3]Climate and Atmosphere Research Center, The Cyprus Institute, Nicosia, Cyprus

**Correspondence:** Nic Surawski (Nicholas.Surawski@uts.edu.au)

**Abstract**

In this study, we use an earth system model with detailed atmospheric chemistry (EMAC v2.55.2) to undertake simulations of hydrogen ($H_2$) atmospheric dynamics. Extensive global equilibrium simulations were performed with a horizontal resolution of 1.9 degrees. The results of this simulation are compared with observational data from 56 stations in the National Oceanic and

Atmospheric Administration (NOAA) Global Monitoring Laboratory (GML) Carbon Cycle Cooperative Global Air Sampling Network. We introduced $H_2$ sources and sinks, the latter inclusive of a soil uptake scheme, that accounts for bacterial consumption. The model thus accounts for detailed $H_2$ and methane ($CH_4$) flux boundary conditions. Results from the EMAC model are accurate and predict the magnitude, amplitude and interhemispheric seasonality of the annual $H_2$ cycle at most observational stations. Time series comparison of EMAC and observational data produces Pearson correlation coefficients ($r$) in excess of

0.9 at eight remote stations located in polar regions and on high mid-latitude islands. A further 23 stations yielded correlation coefficients between 0.7–0.9, predominantly located in remote marine stations across all latitudes and also in polar regions. The quality of model predictions ($r < 0.5$, 9 stations) is reduced in anthropogenically highly polluted stations in east Asia and the Mediterranean region and stations impacted by peat fire emissions in Indonesia, as local and incidental emissions are difficult to capture. Our $H_2$ budget corroborates bottom-up estimates in the literature in terms of source and sink strengths and

overall atmospheric burden. By simulating hydroxyl radicals (OH) in the atmosphere leading to a $CH_4$ lifetime in agreement with observationally constrained estimates, we show that the EMAC model is a capable tool for undertaking high accuracy simulation of $H_2$ at global scale. Future research applications could target the impact of potentially significant natural and anthropogenic $H_2$ sources on air quality and climate, reducing uncertainties in the $H_2$ soil sink and impacts of $H_2$ release on the future oxidising capacity of the atmosphere.

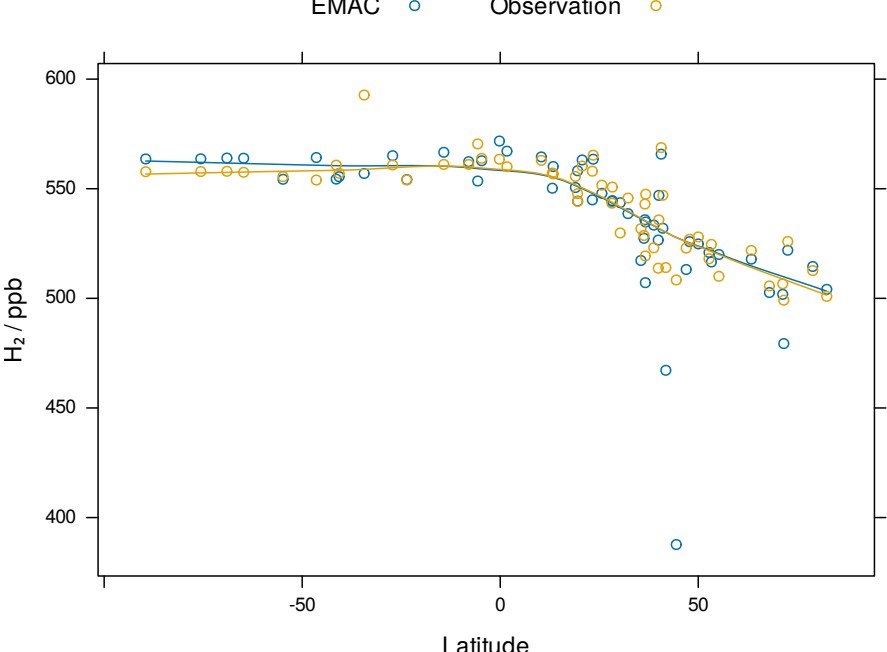

**Key figure**

# 1 Introduction

$H_2$ represents an essential energy vector for 2050 net zero decarbonisation targets to be met. Current demand for $H_2$ equates to approximately 95 Mt per year with existing uses in the refining industry, as well as the chemical industry for production of ammonia, methanol and other chemicals (Hydrogen Council, 2021; International Energy Agency, 2023). $H_2$ is also used in the direct reduction of iron along with smaller uses in electronics, glassmaking and metal processing (International Energy Agency, 2023). With increased governmental, financial and policy support, demand for $H_2$ is forecasted to rise to between 430–690 Mt by 2050 (Hydrogen Council, 2021; International Energy Agency, 2023). Achieving this projected level of hydrogen demand can support clean energy use in: 1) hard to abate sectors such as long-haul trucking, shipping and aviation, 2) sectors that require a clean molecule as a chemical feedstock such as for co-firing of natural gas turbines or industrial processes such as steel manufacturing, 3) sectors that require a source of low carbon heat such as for cement and aluminium production or for buildings (Hydrogen Council, 2021; International Energy Agency, 2023). Use of $H_2$ offers a lot of potential for securing decarbonisation outcomes, provided clean production pathways are prioritised (Hydrogen Council, 2021; International Energy Agency, 2023), carbon capture and storage technologies (if required) work efficiently and at scale (International Energy Agency, 2020), and leakage rates in the $H_2$ value chain are minimised with sound engineering design (Esquivel-Elizondo et al., 2023; Fan et al., 2022).

Despite these potential advantages for decarbonisation, $H_2$ has well-documented climate impacts following its release into the atmosphere which represents an important environmental challenge. In terms of climate impacts, $H_2$ is an indirect greenhouse gas that leads to increases in radiatively active species by increasing 1) $CH_4$ lifetime due to $H_2$ competing for the OH sink 2) tropospheric ozone production due to a chain of reactions initiated by the H atom and 3) stratospheric water vapour that enhances radiative forcing (Derwent et al., 2006; Paulot et al., 2021; Ocko and Hamburg, 2022; Warwick et al., 2022, 2023). Since $H_2$ release affects the oxidising capacity of the atmosphere, it may also lead to changes in the production of sulphate, nitrate and secondary organic aerosols (Sand et al., 2023). Arising from these modelled results, coupled chemistry-climate modelling has a vital role to play before future $H_2$ infrastructure is installed to ensure that projected increases in $H_2$ utilisation do not lead to significant adverse consequences for the earth's atmosphere, air quality and climate.

Simulation of $H_2$ atmospheric chemistry impacts has attracted significant research attention both in the past few decades (Hauglustaine and Ehhalt, 2002; Schultz et al., 2003; Tromp et al., 2003; Warwick et al., 2004) and at present (Derwent et al., 2020; Paulot et al., 2021, 2024; Warwick et al., 2023) given the likelihood that demand for $H_2$ usage will grow and potential environmental impacts still require a solution. Previous attempts at simulating hydrogen mixing ratios with coupled chemistry-climate modelling have met variable levels of success at global scale. In this article, we show that the EMAC model is a highly capable tool for capturing 1) the magnitude, amplitude and seasonality of the annual $H_2$ cycle and 2) the meridional gradients in $H_2$ mixing ratios. These findings support the conclusion that the EMAC model consistently represents the interplay between the dominating soil sink (i.e. 75% of all sink terms) and atmospheric photochemical production (i.e. 63% of all source terms) which is by far the largest source term for $H_2$ (Table 2).

## 2 Materials and methods

In this work, we employ the EMAC model which couples the 5th generation European Centre Hamburg General Circulation Model (ECHAM5; Roeckner et al. (2003, 2004, 2006)) to the Modular Earth Submodel System (MESSy) (Jöckel et al., 2006, 2010). Simulations were performed with T63 spectral resolution which produces a spatial resolution of $1.9°$ (approximately 180–190 km). Simulations were performed with 90 levels up to 80 km above the earth's surface, encompassing both the lower and middle atmosphere. Chemical reactions in the atmosphere were modelled with version 1 of the Mainz Isoprene Mechanism (MIM1; Pöschl et al. (2000); Jöckel et al. (2006)). The model experiment is representative of the present-day (i.e. the year 2020) and uses meteorology for the years 2006-2023, with the first four years used as spin-up time. Flux boundary conditions were employed for both $CH_4$ and $H_2$ to overcome issues with the introduction of artificial sources and sinks arising from using Dirichlet boundary conditions with a prescribed mixing ratio at the lower boundary of the atmosphere. $H_2$ and $CH_4$ are chemically coupled and have nearly the same chemical lifetime (Table 1). Both compete for the OH radical as a chemical sink, with OH being by far the dominant sink for atmospheric $CH_4$ (Saunois et al. (2025); see section 4.1 below). Furthermore, atmospheric oxidation of $CH_4$ is the largest source for $H_2$ (Ehhalt and Rohrer, 2009). To adequately simulate such a coupled system, the EMAC model uses flux boundary conditions for sources and sinks of both species. To reach a steady-state for the control simulation, the initial conditions for $CH_4$ and $H_2$ were obtained from a 15 years long simulation, covering the period 1990–2005. $CH_4$ was simulated based on the work of Zimmermann et al. (2020), in which emissions of $CH_4$ and deposition are represented based on the year 2020. Integration of the equations in the simulation uses a time-step of 450 seconds, and, due to the relatively long lifetime of $H_2$, precluding diel variability, instantaneous values are outputted every day.

### 2.1 Emissions

In this work, the goal is to undertake an equilibrium simulation that reaches steady-state mixing ratios representative of present day atmospheric conditions. Therefore, emissions are based on the year 2020, or the closest year prior to 2020, and are repeated for each year which removes any interannual variability. Due to increasing emissions and its long lifetime $CH_4$ is not in a steady state. Therefore an equilibrium simulation is not fully representative of the atmospheric state in 2020.

For the long-lived tracer $CH_4$, the a posteriori emissions and the best combination of the rising-$CH_4$ scenario of Zimmermann et al. (2020) have been applied. In this work, Zimmermann et al. (2020) show that the EMAC model has been efficient in simulating interactive $CH_4$ mixing ratios over the last two decades. Therein, the model results compare quite well with NOAA and The Advanced Global Atmospheric Gases Experiment (AGAGE) stations and measurements from CARIBIC (Civil Aircraft for the Regular Investigation of the Atmosphere Based on an Instrument Container) flight observations (Brenninkmeijer et al., 2007). Twelve emission categories are considered here, namely, wetlands other than bogs (SWA), animals (ANI), landfills (LAN), rice paddies (RIC), gas production (GAS), shale gas drilling (SHA), bogs (BOG), coal mining (COA), including minor natural sources from oceans, other anthropogenic sources, volcanoes, oil production and offshore traffic, oil-related emissions (OIL), biomass burning (BIB), termites (TER), and biofuel combustion (BFC). Only emissions from bogs, rice fields, wetlands other than bogs, and biomass burning are subject to seasonal variability. Most of the emissions are based on the emission

fields of the Global Atmospheric Methane Synthesis (GAMeS) in which processes with similar isotopic characteristics are aggregated into one group (Houweling et al., 1999). For biomass burning, the GAMeS dataset is replaced by the GFEDv4s (Randerson et al., 2017) and is vertically distributed according to a profile suggested in the EDGAR database (Van Aardenne et al., 2005). The GFEDv4s biomass burning statistics include agricultural waste burning events. A total amount of 601.1 Tg $yr^{-1}$ of $CH_4$ is emitted in the model and detailed emissions for each sector can be found in Table 1 and 3 of Zimmermann et al. (2020), which also describe in detail the emission optimisation process.

$H_2$ emissions were taken from the RETRO dataset (Schultz et al., 2008), which was chosen due to its completeness. As for the other sources, we repeated the emissions based on one single year, namely the year 2000. The RETRO database covers the period 1960–2000, and the last year was taken as representative of 2020 emissions, motivated by the stagnation of $H_2$ emissions in the past few decades (Paulot et al., 2021). A global value of 14.3 Tg $yr^{-1}$ for anthropogenic emissions is obtained from the RETRO database, as well as 4.8 Tg $yr^{-1}$ from soil emissions. Biomass burning emissions were obtained from the GFED (Global Fire Emissions Database) database (Giglio et al., 2013), and accounted for 8.35 Tg $yr^{-1}$. As the RETRO oceanic emissions are outside the range of emissions suggested by the literature (Paulot et al., 2024), these emissions were upscaled to 3 Tg $yr^{-1}$ so to be within the suggested range (i.e. between $3 - 6$ Tg $yr^{-1}$). Both the RETRO and GFED databases provide direct estimates of $H_2$ emissions without relying upon an assumed $H_2/CO$ emissions ratio.

For non-GHGs, different emissions were adopted. Anthropogenic sources of short-lived gases are based on CAMS-GLOB-ANTv4.2 and CAMS-GLOB-AIRv1.1 (Granier et al., 2019), and the emissions are estimated with reduction due to lockdowns during the COVID-19 pandemic (Reifenberg et al., 2022). The reduction in the mixing ratio of the OH radical is below $4\%$ for most of the atmosphere, with the exception of the uppermost troposphere and tropopause region, which is due to reduced flight activity. The small impact on OH is foremost confined to mid-latitudes of the northern hemisphere. Biomass burning emissions are calculated online on a daily basis and rely on dry matter burned from observations and fire type (Kaiser et al., 2012). The emission factors for different tracers and fire types are taken from Andreae (2019) and Akagi et al. (2011). The simulation uses a climatology of the aerosol wet surface density to calculate heterogeneous reactions. It is based on the CMIP5 (Climate Model Intercomparison Project Phase 5) emissions climatology for the years 1996–2005 low S scenario (Righi et al., 2013). The aerosol distribution for radiative forcing calculation is the Tanre climatology (Jöckel et al., 2006). The biogenic emissions of organic species have been compiled following Guenther et al. (1995) and are prescribed in the model in an offline manner (Kerkweg et al., 2006), with the exception of biogenic isoprene and terpenes, for which the emissions are calculated online (Kerkweg et al., 2006).

## 2.2 Soil sink implementation

We estimate the soil sink using a two-layer soil model (Yonemura et al., 2000; Ehhalt and Rohrer, 2013a; Paulot et al., 2021). $H_2$ is assumed to diffuse through a dry top layer of soil with no bacterial activity (layer I), which may be covered by an equally inactive layer of snow. In a second layer below the top layer (layer II), the rate of $H_2$ removal by high-affinity hydrogen-oxidising bacteria (Paulot et al., 2021) depends on both soil temperature and moisture. The resulting deposition rate is parameterised by:

$$v_{\mathrm{d}} = \frac{1}{\left(\delta/D_{\mathrm{soil}}(\theta_{\mathrm{w,I}}) + \delta_{\mathrm{snow}}/D_{\mathrm{snow}} + 1/\sqrt{(D_{\mathrm{soil}}(\theta_{\mathrm{w,II}}) \; A \; g(T) \; f(\theta_{\mathrm{w,II}}/\theta_{\mathrm{p}}))}\right)} \; . \tag{1}$$

The first two terms in parenthesis in the denominator of Eq. (1) represent diffusion through the inactive soil layer and the snow layer of thickness $\delta$ and $\delta_{\mathrm{snow}}$, respectively. The diffusivity of $H_2$ in soil is given by Millington and Quirk (1959):

$$D_{\mathrm{soil}}(\theta_{\mathrm{w}}) = ((\theta_{\mathrm{p}} - \theta_{\mathrm{w}})^{3.1}/\theta_{\mathrm{p}}^2)D_{\mathrm{air}}, \tag{2}$$

which depends on the volumetric soil water fraction $\theta_{\mathrm{w}}$ and the volumetric soil pore fraction (i.e. porosity) $\theta_{\mathrm{p}}$. The diffusivity of $H_2$ in snow is given by:

$$D_{\mathrm{snow}} = 0.64 \; D_{\mathrm{air}}, \tag{3}$$

while the diffusivity of $H_2$ in air is given by:

$$D_{\mathrm{air}} = \frac{0.611 \times 1013.25}{p \cdot ((T + 273.15)/273.15)^{1.75}}, \tag{4}$$

where the diffusivity of $H_2$ in air depends on the air temperature $T$ in $^\circ$C and the air pressure $p$ in hPa.

The third term in parenthesis in the denominator of Eq. (1) represents $H_2$ removal in the lower, active layer. The temperature dependence is given by Ehhalt and Rohrer (2011):

$$g(T) = \frac{1}{\left(1 + \exp(-(T - 3.8)/6.7)\right)} + \frac{1}{\left(1 + \exp((T - 62.2)/7.1)\right)} - 1, \tag{5}$$

where $T$ is the soil temperature in $^\circ$C.

The soil moisture dependence in terms of the water saturation $S = \theta_{\mathrm{w}}/\theta_{\mathrm{p}}$ for eolian sand is given by Ehhalt and Rohrer (2011):

$$f_{\mathrm{es}}(S) = 0.00936 \frac{(S - S_{\mathrm{es}}^*)(1 - S)}{S^2 - 0.1715S + 0.03144}, \tag{6}$$

where $S_{\mathrm{es}}^* = 0.02640$ is the minimum level of water saturation required for microbial activity. For loess loam the soil moisture dependency is given by Ehhalt and Rohrer (2011):

$$f_{\mathrm{ll}}(S) = 0.01997 \frac{(S - S_{\mathrm{ll}}^*)(0.8508 - S)}{S^2 - 0.7541S + 0.2806}, \tag{7}$$

where $S_{\mathrm{ll}}^* = 0.05369$. For a mixture of eolian sand and loess loam we use the weighted mean given by:

$$f(S) = \varphi_{\mathrm{sand}} f_{\mathrm{es}}(S) + (1 - \varphi_{\mathrm{sand}})f_{\mathrm{ll}}(S), \tag{8}$$

where $\varphi_{\mathrm{sand}}$ is the sand fraction of the soil.

The resolution-dependent constant $A$ represents bacterial activity and is adjusted to yield a global mean deposition velocity of

$0.033$ cm s$^{-1}$ over land during 2012 to 2015 (Yashiro et al., 2011). Using the $0.25°$ grid spacing of the ERA5 input data, we obtain $A = 10.9$.

     The thickness of the upper soil layer without hydrogenase (i.e. an enzyme in prokaryotes such as bacteria that consume H$_2$) activity is parametrised by:

$$\delta_{\mathrm{s}} = 0.0057((\theta_{\mathrm{p}} - \theta_{\mathrm{w}})/\theta_{\mathrm{w}})^{2.5}, \tag{9}$$

in sandy loam and

$$\delta_{\mathrm{l}} = 0.109((\theta_{\mathrm{p}} - \theta_{\mathrm{w}})/\theta_{\mathrm{w}})^{1.8}, \tag{10}$$

in loam. Both $\delta_{\mathrm{s}}$ and $\delta_{\mathrm{l}}$ are expressed in cm. For a mixture of sandy loam and loam with sand fraction $\varphi_{\mathrm{sand}}$ we use the weighted mean to calculate the soil layer thickness via:

$$\delta = \varphi_{\mathrm{sand}}\delta_{\mathrm{s}} + (1 - \varphi_{\mathrm{sand}})\delta_{\mathrm{l}}. \tag{11}$$

The soil water content in the top, dry layer (i.e. $\theta_{\mathrm{wI}}$) is assumed to be the threshold moisture content below which the bacterial activity vanishes, i.e. there is no H$_2$ uptake in this layer, and is given by:

$$\theta_{\mathrm{wI}} = S^*\theta_{\mathrm{p}}, \tag{12}$$

where $S^* = S^*_{\mathrm{es}}$ is the threshold moisture content for eolian sand and $S^* = S^*_{\mathrm{ll}}$ for loess loam.

     Accordingly, the remaining water within the top 10 cm of soil is between depth $\delta$ and 10 cm, resulting in a soil water content
for the second layer (i.e. $\theta_{\mathrm{wII}}$) of:

$$\theta_{\mathrm{wII}} = \frac{10\theta_{\mathrm{w}} - \delta\theta_{\mathrm{wI}}}{10 - \delta}. \tag{13}$$

     We evaluated Eq. (1) using monthly reanalysis data for soil moisture, soil temperature, air pressure, snow depth and snow density with a $0.25°$ grid spacing from the ERA5 dataset provided by the European Centre for Medium-Range Weather Forecasts (ECMWF) (Hersbach et al., 2020, 2023). The mean soil moisture and soil temperature for the top 10 cm soil layer was
obtained by linearly interpolating the ERA5 soil level data. The volumetric soil water content was then uniformly reduced by 6% (Paulot et al., 2021). Static soil porosity and sand fraction maps with $0.25°$ grid spacing were obtained from the Land Data Assimilation System (LDAS) (Rodell et al., 2004; GLDAS, 2024).

     The global distribution of the H$_2$ deposition velocity averaged over the 2012 to 2021 period resembles the maps presented by Paulot et al. (2021), but with more pronounced extrema (Figure S1 in the supplement). The zonal mean of the H$_2$ deposition
velocity over land also falls within the range of results reported by Paulot et al. (2021). However, the local minimum around 20°N, due to the Sahara, and the maximum around 10°N, where the transition to more humid regions favours soil uptake, are more distinct in this study than in most of those shown by Paulot et al. (2021) (Figure S2 in the supplement). Thanks to extensive observational records of atmospheric hydrogen concentrations at stations around the world, it is possible to carry out detailed validation of hydrogen deposition based on the resulting concentrations. This method, presented in the following
sections, is more informative than local deposition analysis because it is less susceptible to significant variations in surface conditions and is based on a much larger database.

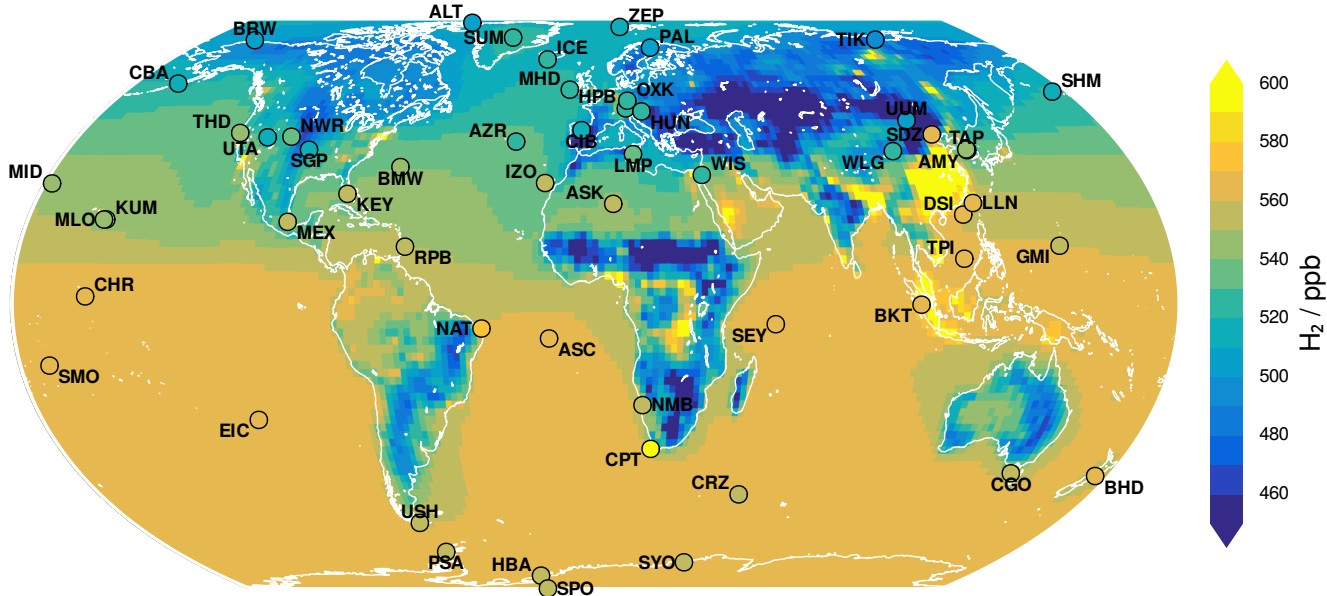

**Figure 1.** Global map of H$_2$ mixing ratios and the location of observational stations. Model data is averaged over the years 2010–2023 (inclusive) which is representative of the year 2020 for a steady-state simulation, while observational data uses mid-2020 values from a detrending fit using a sixth order harmonic regression technique.

## 2.3 Observations

EMAC simulations are compared with observational data from 56 stations (with more than 12 monthly values) that form part of the NOAA GML Carbon Cycle Cooperative Global Air Sampling Network (Petron et al., 2024). Data gaps exist at some stations due to the application of quality control procedures, as well as missing data due to impacts of the COVID-19 pandemic. For comparison with the results of the EMAC equilibrium simulation, the observed monthly values have been detrended by subtracting the trend obtained by a sixth order harmonic regression with a linear trend term, while keeping the mid-year values for 2020 fixed.

## 3 Results

Figure 1 presents a global map of modelled annual mean H$_2$ mixing ratios as well as the location of observational stations that we use for model inter-comparison purposes. This global map shows the inter-hemispheric gradient for this molecule whereby H$_2$ mixing ratios are higher in the southern hemisphere compared to the northern hemisphere. This global map also shows the influence of pollution hotspots in Asia, and the influence of biomass burning emissions in central Africa and peat fire emissions in southeast Asia.

We show time series data for model comparisons with observational data (without gaps in monthly data) from 20 observational stations in Figure 2. Further comparisons with observational data from another 36 observational stations (with data gaps) are shown in Figures B1 and B2. 38 of the 56 stations have marine characteristics (Table A1 and Figure 1), i.e. are located coastally or on islands. 11 stations can be described as polar (latitude $\geq |60°|$), and 14 stations are positioned on mountains (i.e. a single station might have several characteristics). These stations usually measure remote, often marine air masses or free tropospheric concentrations. Just 8 stations Hungary (HUN), Mongolia (UUM) , CIBA (CIB, northern central Spain), Shangdianzi (SDZ, China), Wendover (UTA, Utah), Southern Great Plane (SGP, Oklahoma), Israel (WIS), and Gobabeb (NMB, Namibia) fall in none of these previous categories. Inner continental non-mountain stations are sparse, with none in South America and Australia. The two continental stations in Africa (ASK, and NMB) are located in the desert. There are also no monitoring stations in Central Asia. The three Mediterranean stations (CIB, LMP, and WIS) and the continental Chinese (SDZ) and Mongolian (UUM) stations are located in or close to areas with high air pollution levels (Lelieveld et al. (2002); Silver et al. (2025)). The Bukit station (BKT, Indonesia) is also regularly impacted by biomass burning and peat fires Yokelson et al. (2022).

Across these 56 observational stations, the Pearson correlation coefficient ($r$) exceeds 0.9 for eight remote stations located either in the Arctic or Antarctic regions and on islands at high mid-latitudes. In such cases, the annual cycle of $H_2$ is modelled excellently in terms of magnitude, amplitude and seasonality. In contrast, nine stations produce correlation coefficients below 0.5. The three stations in the Mediterranean region CIB (-0.17), LMP (0.05) and WIS (-0.29) and the Chinese and the Mongolian stations SDZ (-0.2) and UUM (0.03) show correlation coefficients close to zero or even negative. Negative $r$ values suggest that the EMAC model does not correctly capture the phasing of the annual $H_2$ cycle which results in pronounced phase mismatches or anti-correlation in Table B1 and Figures 2, B1, and B2. These stations are located in regions known for high levels of air pollution. The two tropical coastal stations BKT (0.14) and the Natal station in Brazil (NAT, 0.14) do show rather low $r$ values compared to other tropical stations. They are impacted by biomass burning (i.e. NAT) and peat fire emissions (i.e. BKT). The comparison at the Hungarian station (HUN, 0.39) shows a considerably lower $r$ value compared to the other two central European stations Hohenpeissenberg (HPB, 0.56) and Ochsenkopf (OXK, 0.64). HPB and OXK are mountain stations less susceptible to local mismatches in $H_2$ soil deposition, as can be seen from the much better matching of model amplitude and phasing with measurements. The Christmas Island station (CHR, 0.43) is the only very remote tropical island station, which has a considerably low $r$ value. At this station, the observational time series is relatively short with hardly any annual cycle. Another 23 stations have correlation coefficients between 0.7–0.9 which demonstrate very good agreement between model and observational data. A number of these stations are located in the mid-latitudes either in the northern or southern hemisphere. The remaining 16 stations produce correlation coefficients between 0.5–0.7 mostly in either remote tropical or mid-latitude regions. Especially the Antarctic stations show an upward trend for atmospheric $H_2$. The model with its emissions and soil sink repeating the year 2020, cannot capture this feature. This trend coincides with further increasing atmospheric $CH_4$ concentrations after 2010 following its hiatus of the previous decade (Lan et al., 2024). Table 2 shows that oxidation of atmospheric $CH_4$ is the largest source term for $H_2$ Ehhalt and Rohrer (2009). To investigate this further in the future, a model simulation with flux boundary conditions for $H_2$ and $CH_4$ in transient mode is needed. Overall, the results are very promising

and demonstrate the ability of the EMAC model to predict $H_2$ mixing ratios accurately in most regions of the earth. To provide a visual overview of these results, Figure 3 provides a global map of Pearson correlation coefficients for comparison of EMAC and observational data.

We also present a plot of the meridional gradient in $H_2$ in Figure 4. Overall, meridional gradients in $H_2$ are captured very well by the EMAC model, notably for stations located in the southern hemisphere, likely because many represent the background atmosphere, whereas many stations in the northern hemisphere are affected by local influences. The model correctly predicts higher $H_2$ mixing ratios in the southern hemisphere even though the majority of $H_2$ sources are present in the northern hemisphere. The predicted interhemispheric gradient in $H_2$ presented here is correct by virtue of the greater soil sink that is present in the northern hemisphere arising from its larger land area (Ehhalt and Rohrer, 2009). Most of the discrepancies between the observed and predicted $H_2$ mixing ratios exist for a small number of stations within the northern hemisphere mid-latitudes (between 30–60° N) and in the tropics, presumably influenced by local source variability that is insufficiently resolved by our global model. The coverage of many continental land masses by the observational stations is sparse. For South America, Australia, Africa, Central Asia, Siberia and India there are almost no measurements available. This is a problem, especially in validating the soil sink, which can be considered the most uncertain part of the $H_2$ budget (Paulot et al., 2021).

For further results, we refer the reader to Appendix B which provides further graphs and tabulated summaries of model performance.

## 4  Discussion

### 4.1  Model comparison with observational data

A key feature of the results (Figures 2, B1, and B2) is the ability of the EMAC model to realistically predict the magnitude, amplitude and seasonality of the annual $H_2$ cycle at most stations, in unison with that from $CH_4$ Zimmermann et al. (2020), with both compounds being modelled with flux boundary conditions and interactive sinks. Promising results are obtained especially for stations that experience remote air masses, for example, in mostly polar regions, which are particularly sensitive to atmospheric transport and chemistry dynamics. The EMAC model results are also quite promising in a range of mid-latitude stations both in the northern and southern hemisphere. In contrast, there are some regions of the globe (Figure 3) where results are not as promising. For example, the EMAC model predictions are less accurate in the highly anthropogenically polluted Mediterranean region, near the Amazonian region which is impacted by biomass burning emissions, and southeast Asia which is impacted by peat fire emissions. Due to the coarse spatial resolution of 180–190 km and limited information about local and incidental sources, the variability of mixing ratios in these regions is more challenging to capture. This is especially the case for some coastal stations (e.g. NAT, BKT) where the model is limited due to resolution in accurately representing the mixing of marine and continental air. Also the deviations in China and Mongolia (SDZ, UUM) can be partly attributed to a resolution effect. Both stations are located close to strong horizontal gradients in $H_2$ mixing ratios. The vertical resolution of the lowermost model layers (i.e. thicknesses of 66 m, 166 m and 319 m from the surface upwards) and the representation of the orography influence the comparison. It is important to consider the measurement height relative to the surface and the

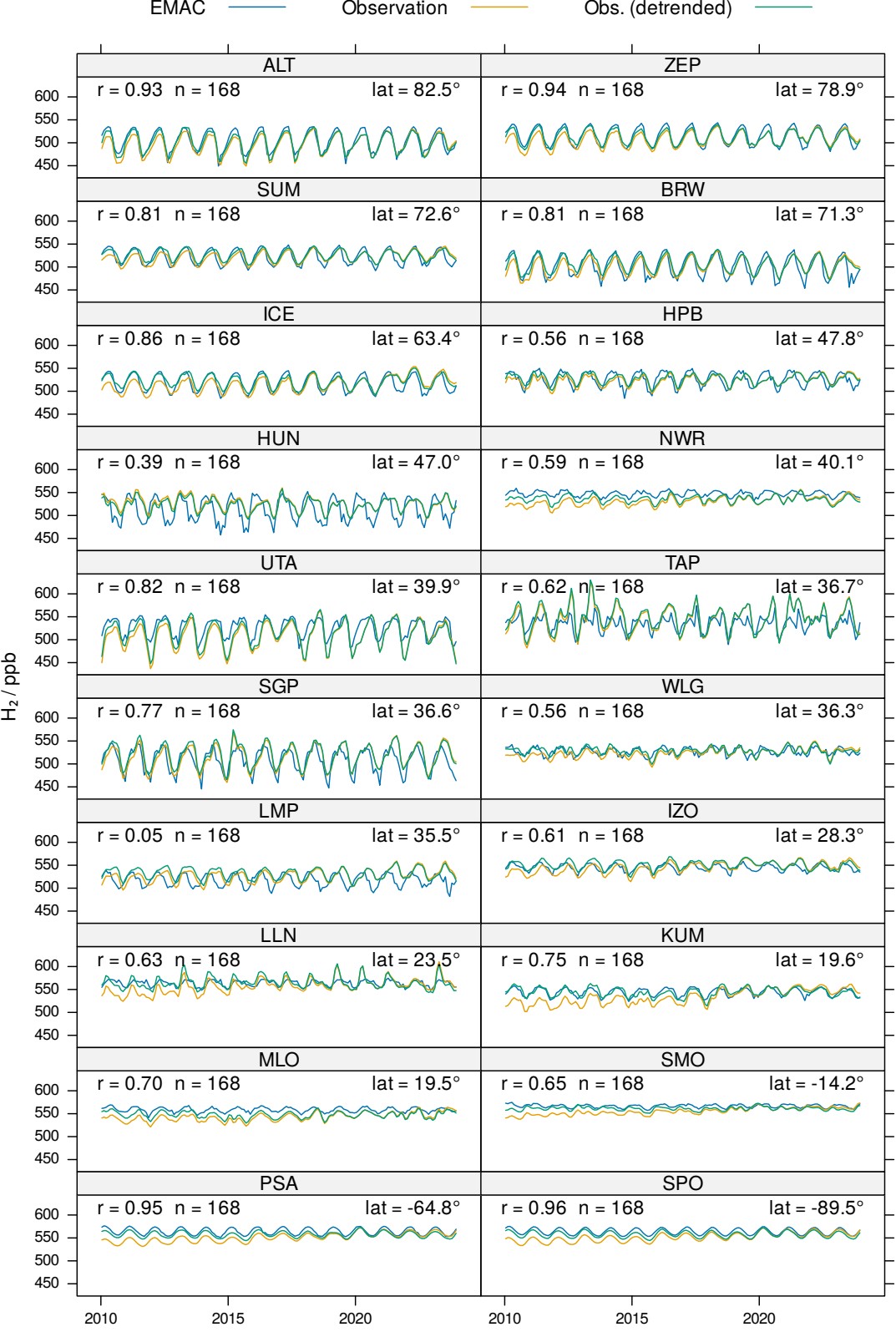

**Figure 2.** Time series comparison of observational and EMAC model data for $H_2$. Results are presented for 20 stations without any gap in monthly data. The sample size for observational data is denoted by $n$, while $r$ is the Pearson correlation coefficient. Latitudes are denoted by lat.

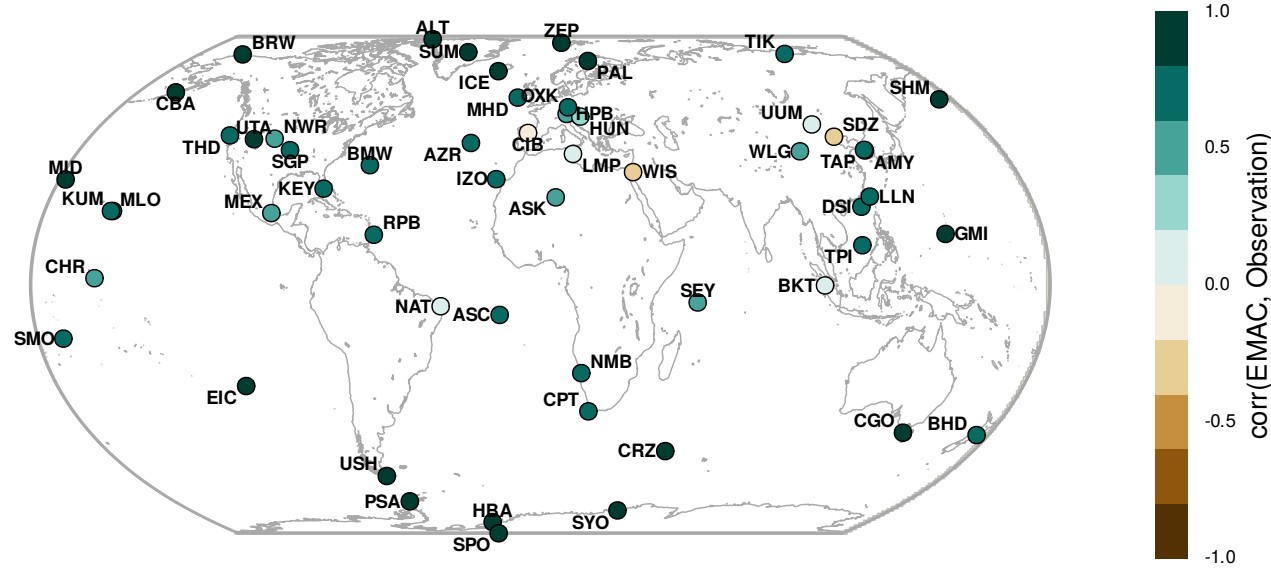

**Figure 3.** Pearson correlation coefficient for the intercomparison between EMAC and observational data for $H_2$. Model data is compared with detrended observational data for the years 2010–2023 (inclusive) to perform this calculation.

geographic prominence of the stations. For example, the modelled amplitude of the annual $H_2$ cycle can be reduced by up to 40% between the surface and the next model layer for continental stations due to the importance of the soil sink, whereas the

$H_2$ mixing ratios increase with height driven by the strong atmospheric chemical $H_2$ production. In addition, an interesting model-measurement discrepancy occurs at the Weizmann Institute of Science (WIS) station near the northern Red Sea, where unaccounted for alkane emissions have been attributed to natural seepage from deep water sources (Bourtsoukidis et al., 2020), possibly accompanied by $H_2$ emissions. Overall, the EMAC model performs favourably at a global scale for simulating $H_2$ mixing ratios. Comparison with model output from Yashiro et al. (2011) shows that while the EMAC model produces

correlation coefficients in excess of 0.7 for over half of the observational stations, the CHASER chemistry-climate model achieves the same result for only one quarter of all observational stations. The annual mean $H_2$ mixing ratios are well captured by the EMAC model (Table B1 and Figure 4), with the exception of CIB ($r$=-0.17), UUM ($r$=0.03), and the coastal station Cape Town (CPT) despite its high Pearson correlation coefficient ($r$=0.73). As mentioned above, the $H_2$ observational stations do not represent the inner region of continents well. Especially, the small number (i.e. UTA, SGP, UUM) of remote non-mountain,

non-hyperarid stations, placed away from highly anthropogenically influenced regions limits its usage in looking in more detail on the parametrisation of the soil sink. At Wendover (UTA, Utah, arid cold climate, $r = 0.82$) and the Southern Great Planes (SGP, Oklahoma, humid subtropical climate, $r = 0.77$), shown in Figure 2, the model performs well by realistically representing the magnitude, amplitude and seasonality of the observations, indicating that the soil sink parametrisation of $H_2$

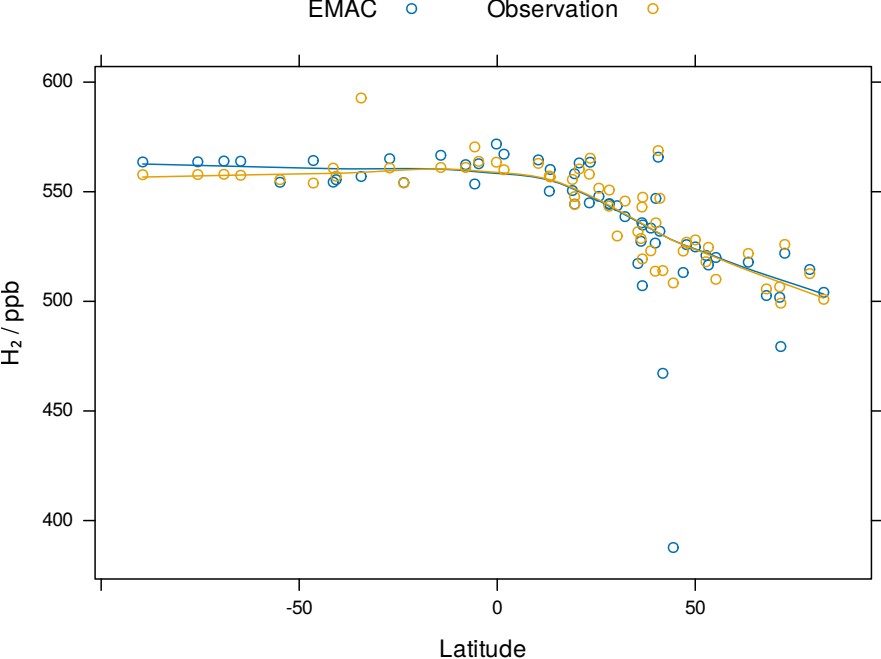

**Figure 4.** Meridional gradients in $H_2$ for EMAC predictions and observational data where stations had more than 12 monthly values. The solid lines were obtained by locally estimated scatterplot smoothing (LOESS; smoothing parameter 2 / 3, locally linear). Negative latitudes represent south and positive latitudes represent north of the equator. Data are shown for stations with more than 12 monthly values.

is realistic in those regions. The poor results at the Mongolian station (UUM, arid cold climate, $r = 0.03$), Figure B1, might be
partly attributed to a resolution effect (see above).

To successfully simulate $H_2$ mixing ratios in the atmosphere, a model needs to correctly resolve the complex interplay between meteorology and chemistry. In terms of chemistry, having the oxidising capacity of the atmosphere represented correctly is a key consideration (Prather and Zhu, 2024). It is largely controlled by the concentration of OH radicals in the troposphere (Lelieveld et al., 2016) and determines the atmospheric lifetime of numerous species including $CH_4$. The total $CH_4$ sink is
largely dominated by its reaction with OH (see Saunois et al. (2025) for a review). In this sense, $CH_4$ lifetime is a measure for the total oxidative capacity of the atmosphere. Observational estimates derived from methyl chloroform ($CH_3CCl_3$) measurements lead to a total atmospheric $CH_4$ lifetime of $9.1 \pm 0.9$ years (Prather et al., 2012). Our estimate of $9.3 \pm 0.06$ years (Table 1) compares well, and is only marginally higher than the range indicated by Prather et al. (2012). This also holds for the global tropospheric chemical $CH_4$ sink. The supporting information of Prather et al. (2012) states that for the tropospheric
$CH_4$ lifetime based on the reaction with OH (i.e. $11.2 \pm 1.3$ years) and for the lifetime of $CH_4$ based on the reaction with tropospheric chlorine (i.e. $200 \pm 100$ years) yields a combined tropospheric chemical lifetime of $10.6 \pm 1.2$ years for $CH_4$. This value compares quite well with our estimate of 10.4 years (Table 1). Model intercomparisons performed by Nicely et al. (2020) suggests that many chemistry models underestimate $CH_4$ lifetime due to simulating an atmosphere that is overly enriched in

OH radicals. Recently, work by Yang et al. (2025) concurs that several atmospheric chemistry models over-predict OH mixing ratios which has implications for $CH_4$ and $H_2$ lifetimes. The multi-model estimate from CMIP6 (Coupled-Model Intercomparison Project 6; Collins et al. (2017)) and CCMI (Chemistry Climate Model Initiative; Plummer et al. (2021)) models used by the Global Carbon Project community for the bottom-up estimates of $CH_4$ sources yields a total $CH_4$ lifetime of 8.2 years with a range of 6.8–9.7 years (Saunois et al., 2025). It also shows a large spread, which propagates into increased uncertainties for the derivation of emission budgets. We believe that the EMAC model is realistically capturing the total oxidising capacity of the atmosphere which helps to facilitate high-accuracy prediction of $H_2$ and $CH_4$ dynamics.

## 4.2 Budget and lifetimes

In the atmosphere, $CH_4$ and $H_2$ are tracers strongly connected with similar chemical fates. Table 1 shows that $CH_4$ and $H_2$ have nearly identical chemical lifetimes both in the troposphere and atmosphere. Furthermore the total sources of $H_2$ and $CH_4$, corrected for molecular masses, are very comparable. The biggest difference between these two compounds stems from hydrogen's much larger soil sink which reduces its tropospheric lifetime by approximately a factor of four compared to $CH_4$.

In Table 2, we compare our $H_2$ budget derived from EMAC model output with other estimates from the literature. We find that our $H_2$ budget agrees favourably with bottom-up literature estimates that rely on a combination of emission datasets and model calculations of turnovers and loss rates, but differs from top-down estimates relying on either inverse modelling (Xiao et al., 2007) or analysis of the $^2H$ (i.e. deuterium) budget (Rhee et al., 2006). Our overall budgeting of sources and sinks agrees very well with bottom-up estimates. In addition, our tropospheric $H_2$ lifetime is in very good agreement with bottom-up estimates. The tropospheric burden is in the upper range of model estimates. Note, that the upper boundary of the tropospheric range is often not clearly defined in the literature, with different definitions e.g. 100 hPa, World Meteorological Organization (WMO), or a climatological tropopause being used. In this study the WMO tropopause definition is used based on a dynamic tropopause in high latitudes and lapse rate being used at low latitudes. The photochemical production is in between the range for bottom-up and top-down estimates (Paulot et al., 2021). These findings suggest that the EMAC model simulates a realistic atmospheric oxidation capacity which is a critical requirement for predicting $H_2$ mixing ratios well.

Recent work by the United States Geological Survey (Ellis and Gelman, 2024) has developed a simple, zero-dimensional mass balance model coupled with Monte Carlo uncertainty analysis to explore global potential for geological (or gold) $H_2$ production in the earth's crust. Median modelled estimates of the subsurface $H_2$ resource are approximately $5.6 \times 10^6$ megatonnes. Ellis and Gelman (2024) estimate that global geological $H_2$ resources cause an additional global flux of 24 Tg yr$^{-1}$ from the subsurface to the atmosphere. This is speculative and would add unaccounted $H_2$ emissions almost of the strength of the current non-photochemical sources. Current knowledge concerning the budget of atmospheric $H_2$ does not exclude the existence of a large geological $H_2$ reservoir, and further emphasises the importance of dry deposition for the global atmospheric $H_2$ budget.

**Table 1.** Chemical budgets and lifetimes for $H_2$ and $CH_4$. Uncertainties are calculated as the standard deviation of multi-year annual global means. Note that lifetimes are always calculated with respect to global burden (Prather et al., 2012; SPARC, 2013).

| Budget term | $H_2$ | $CH_4$ |
|---|---|---|
| Tropospheric chemical sink (Tg yr$^{-1}$) | $19.0 \pm 0.16$ | $534.5 \pm 4.04$ |
| Tropospheric chemical production (Tg yr$^{-1}$) | $49.5 \pm 0.43$ | - |
| Tropospheric chemical lifetime (years) | $10.5 \pm 0.08$ | $10.4 \pm 0.08$ |
| Atmospheric chemical lifetime (years) | $9.6 \pm 0.07$ | $9.8 \pm 0.07$ |
| Soil sink (Tg yr$^{-1}$) | $60.5 \pm 0.07$ | $30.9 \pm 0.02$ |
| Tropospheric lifetime (years) | $2.5 \pm 0.004$ | $9.9 \pm 0.07$ |
| Atmospheric lifetime (years) | $2.5 \pm 0.004$ | $9.3 \pm 0.06$ |

### 4.3 Suggestions for future applications

Future research efforts in modelling $H_2$ atmospheric chemistry could build on the current work in three key ways. Firstly, scenarios could be constructed to explore what role geological $H_2$ (i.e. gold $H_2$) holds for future atmospheric chemistry. If economically extractable reserves of gold $H_2$ are found, future utilisation of $H_2$ would increase well beyond current projections (Hand, 2023; Truche et al., 2024; Ellis and Gelman, 2024). It would therefore be critical to assess the atmospheric chemistry implications of vastly increased $H_2$ usage. Secondly, it will be critical to assess what impact $H_2$ use has on the future oxidising capacity of the atmosphere. Clean $H_2$ use will be associated with significant reductions in the co-emission of criteria pollutants (Galimova et al., 2022) which will influence the formation of atmospheric oxidants such as ozone and OH radicals that constrain $CH_4$ and $H_2$ lifetimes (Archibald et al., 2011; Brasseur et al., 1998; Ganzeveld et al., 2010). Thirdly, the $H_2$ budget is dominated by the land sink (Tables 1 and 2) and future research efforts could help to constrain the important role played by a number of soil properties (e.g. porosity, soil moisture, temperature, and organic carbon content) on terrestrial $H_2$ uptake (Ehhalt and Rohrer, 2011, 2013a; Paulot et al., 2021; Smith-Downey et al., 2006). The production of $H_2$ by enzymes in soil (i.e. hydrogenases) could also be considered in a depth-resolved manner as knowledge of the underlying processes improves (Ehhalt and Rohrer, 2013b). Recent coupling of the JSBACH vegetation model to EMAC by Martin et al. (2024) has developed a potential model tool for undertaking on-line $H_2$ land sink calculations.

**Table 2.** Tabulation of the $H_2$ budget from this study and from literature estimates. Uncertainties are calculated as the standard deviation of multi-year annual means.

| Budget term | This study (EMAC) | Seiler and Conrad (1987) | Warneck (1988) | Novelli et al. (1999) | Hauglustaine and Ehhalt (2002) | Sanderson et al. (2003) | Rhee et al. (2006) | Price et al. (2007) | Xiao et al. (2007) | Ehhalt and Rohrer (2009) | Pieterse et al. (2011) | Yashiro et al. (2011) | Paulot et al. (2021) | Sand et al. (2023)[c] | Paulot et al. (2024) |
|---|---|---|---|---|---|---|---|---|---|---|---|---|---|---|---|
| **Sources: Tg yr$^{-1}$** | | | | | | | | | | | | | | | |
| **Tropospheric** | 79.1 ± 0.4 | 87 ± 38 | 89 | 77 ± 16 | 70 | 78.2 | 107 ± 11 | 73 | | 76 ± 14 | 77.3 | 73–80 | 74.4 | 74–102 | 74 ± 1 |
| Photochemical | 49.5 ± 0.4 | 40 ± 15 | 50 | 40 ± 16 | 31 | 30.2 | 64 ± 12 | 34.3 | | 41 ± 11 | 37.3 | 38–39 | 42.1 | 34–56 | 44 |
| CH$_4$ oxidation | 34.5 ± 0.4 | 15 ± 5 | 29 | 26 ± 9 | | 15.2 | | 24.5 | | 23 ± 8 | | | | | 27 |
| VOC oxidation | 15.0 ± 0.2 | 25 ± 10 | 21 | 14 ± 7 | | 15 | | 9.8 | | 18 ± 7 | | | | | 17 |
| Direct | 29.6 | 47 | 39 | 37 | 39 | 48 | 43 | 38.8 | 27 | 35 | 40 | 30–37 | 32.3 | 23–68 | 30 |
| Ocean | 3$^a$ | 4 ± 2 | 4 | 3 ± 2 | 5 | 4 | 6 ± 5 | 6 ± 3 | | 6 ± 3 | 5 | 6 | 6 | | 6 |
| Biofuel | | | | | | | | 4.4 | | | | | | | |
| Soil | 4.8 | 3 ± 2 | 3 | 3 ± 1 | 5 | 4 | 6 ± 5 | 0 | | 3 ± 2 | 3 | 3 | 3 | | 3 |
| Biomass burning | 7.5 | 20 ± 10 | 15 | 16 ± 5 | 13 | 20 | 16 ± 3 | 10.1 | 12 ± 3 | 15 ± 6 | 15 | 8–15 | 9 | | 8 |
| Anthropogenic | 14.3 | 20 ± 10 | 17 | 15 ± 10 | 16 | 20 | 15 ± 6 | 18.3 | 15 ± 10 | 11 ± 4 | 17 | 15.1–15.4 | 14.3 | | 13 |
| **Atmospheric** | 81.1 ± 0.4 | | | | | | | | 103 ± 10 | | | | | | |
| Photochemical | 51.5 ± 0.4 | | | | | | | | 76 ± 9 | | | | | | |
| Stratospheric CH$_4$ oxidation | 1.94 ± 0.02 | | | | | | | | | | | | | | |
| Stratospheric VOC oxidation | 0.08 ± 0.02 | | | | | | | | | | | | | | |
| **Sinks: Tg yr$^{-1}$** | 81.2 ± 0.2 | 98 ± 23 | 89 | 75 ± 41 | 70 | 75.4 | 107 ± 11 | 73 | 103.9 | 79 | 77.9 | 75–78 | 75.1 | 75–102 | |
| Soil uptake | 60.5$^b$ ± 0.1 | 90 ± 20 | 78 | 56 ± 41 | 55 | 58.3 | 88 ± 11 | 55 ± 8.3 | 84 ± 8 | 60 | 55.8 | 57–60 ± 12 | 54.7 | 44–73 | |
| Photochemical | 20.8 ± 0.2 | 8 ± 3 | 11 | 19 ± 5 | 15 | 17.1 | 19 ± 3 | 18 | 19.9 | 19 | 22.1 | 17–18 | 20.4 | 22–30 | |
| Troposphere | 19.0 ± 0.2 | | | | | | | | 18 ± 3 | | | | | | |
| Stratosphere | 1.8 ± 0.01 | | | | | | | | 1.9 ± 0.3 | | | | | | |
| **Burden: Tg** | 199.6 ± 0.2 | | | | | | | | 191 ± 29 | | | | | 184–209 | |
| Stratosphere | 34.4 ± 0.1 | | | | | | | | 42 | | | | | | |
| Troposphere | 165.2 ± 0.3 | | 163 | 155 ± 10 | 136 | 172 | 150 | 141 | 149 ± 23 | 155 ± 10 | 169 | 148–153 | 157.4 | | |
| IHD (ppbv) | 29.4 ± 0.4 | | | | | | | | | | | | | | |
| Lifetime (years) | 2.1 ± 0.003 | | 1.8 | 2.1 | 1.9 | 2.2 | 1.4 | 1.9 | 1.4 ± 0.2 | 2 | 2.2 | 1.9–2.0 | 2.1 | 1.9–2.7 | 2.5 |

$^a$ Up-scaled from 0.5 to 3 Tg/yr to match literature recommendations (Paulot et al., 2021); $^b$ the dry deposition velocity of $H_2$ Paulot et al. (2021) has been reduced by 6% (from the continental global mean of 0.035 to 0.033 cms$^{-1}$ to improve simulated $H_2$ especially in polar latitudes. $^c$ Multi-model results are presented as a range. VOC = volatile organic compound. IHD = interhemispheric difference.

## 5  Conclusions

In this study, we have successfully extended and used the EMAC model to undertake simulations of $H_2$ atmospheric dynamics, constrained by flux boundary conditions for both $H_2$ and $CH_4$. Comparing the EMAC model output with observational data at 56 stations from the NOAA GML Carbon Cycle Cooperative Global Air Sampling Network generally indicates very good agreement at global scale. Excellent results are achieved at remote observational stations and for stations measuring remote and free tropospheric air, suggesting that atmospheric source, sink and transport processes are accurately represented, while model performance is degraded at stations impacted by nearby pollution sources. Our $H_2$ budget is also in good agreement with bottom-up estimates in the literature. We find that the EMAC model simulates the $CH_4$ chemical lifetime in excellent agreement

with observational estimates, which suggests the model calculates OH radical mixing ratios in a representative manner. The $H_2$ soil sink, based on a two-layer soil model (Yonemura et al., 2000; Ehhalt and Rohrer, 2013a; Paulot et al., 2021), in combination with monthly ERA5 reanalysis data for soil related parameters has been successfully used by the EMAC model. We conclude that atmosphere chemistry models with such features, capturing the most dominant terms of the atmospheric $H_2$ budget, should be able to generally simulate station observations of atmospheric hydrogen. This gives confidence that scenario simulations regarding the future $H_2$ economy will provide reliable estimates of its atmospheric impact.

*Code and data availability.* The Modular Earth Submodel System (MESSy) is in continuous development and is used by a consortium of institutions. Source code access and usage is licensed to all affiliates of institutions which are members of the MESSy Consortium. Institutions can become a member of the MESSy Consortium by signing the MESSy Memorandum of Understanding. The MESSy Consortium website (http://www.messy-interface.org) provides further information regarding access to the model. The exact version of the EMAC v2.55.2 source code and simulation set-ups used to produce the results used in this paper is archived on the Zenodo repository at https://doi.org/10.5281/zenodo.15211346 (The MESSy Consortium, 2025).

Regarding data availability, access to the NOAA GML Carbon Cycle Cooperative Global Air Sampling Network data is available at https://doi.org/10.15138/WP0W-EZ08 (Petron et al., 2024), the ERA5 reanalysis data is available at https://doi.org/10.24381/cds.adbb2d47 (Hersbach et al., 2023), and the Global Fire Emissions Database (GFED) v4.1 data is available at https://doi.org/10.3334/ORNLDAAC/1293 (Randerson et al., 2017). The monthly $H_2$ deposition velocity at $0.25°$ resolution is available from the Edmond Open Research Data Repository of the Max Planck Society (Klingmüller, 2025).

**Appendix A:  List of observational stations**

**Table A1.** Observational Stations from the NOAA GML Carbon Cycle Cooperative Global Air Sampling Network. The station code is complemented with p for polar (latitude ≥ |60°|), ma for marine i.e. coastal/island, and m for mountain for further characterisation.

| Station code | Station Name | Latitude | Longitude | Elevation (masl) | Country | Cooperating Agencies |
|---|---|---|---|---|---|---|
| ALT pma | Alert, Nunavut | 82.4508° North | 62.5072° West | 185 | Canada | Environment Canada |
| AMY ma | Anmyeon-do | 36.5389° North | 126.3295° East | 47 | Republic of Korea | Korea Global Atmosphere Watch Center, Korea Meteorological Administration |
| ASC ma | Ascension Island | 7.9667° South | 14.4° West | 85 | United Kingdom | Met Office (United Kingdom) |
| ASK m | Assekrem | 23.2625° North | 5.6322° East | 2710 | Algeria | Office National de la Meteorologie |
| AZR ma | Terceira Island, Azores | 38.766° North | 27.375° West | 19 | Portugal | Instituto Nacional de Meteorologia e Geofisica |
| BHD ma | Baring Head Station | 41.4083° South | 174.871° East | 85 | New Zealand | National Institute of Water and Atmospheric Research |
| BKT mma | Bukit Kototabang | 0.202° South | 100.318° East | 845 | Indonesia | Bureau of Meteorology and Geophysics |
| BMW ma | Tudor Hill, Bermuda | 32.2647° North | 64.8788° West | 30 | United Kingdom | Bermuda Institute of Ocean Sciences |
| BRW pma | Barrow Atmospheric Baseline Observatory | 71.323° North | 156.6114° West | 11 | United States | NOAA Global Monitoring Laboratory |
| CBA ma | Cold Bay, Alaska | 55.21° North | 162.72° West | 21.34 | United States | U.S. National Weather Service |
| CGO ma | Cape Grim, Tasmania | 40.683° South | 144.69° East | 94 | Australia | CSIRO |
| CHR ma | Christmas Island | 1.7° North | 157.1518° West | 0 | Republic of Kiribati | Dive Kiribati |
| CIB | Centro de Investigacion de la Baja Atmosfera (CIBA) | 41.81° North | 4.93° West | 845 | Spain | Centro de Investigacion de la Baja Atmosfera, University of Valladolid |
| CPT ma | Cape Point | 34.3523° South | 18.4891° East | 230 | South Africa | South African Weather Service |
| CRZ ma | Crozet Island | 46.4337° South | 51.8478° East | 197 | France | Centre des Faibles Radioactivities/TAAF |
| DSI ma | Dongsha Island | 20.6992° North | 116.7297° East | 3 | Taiwan | National Central University, Taiwan |
| EIC ma | Easter Island | 27.1597° South | 109.4284° West | 47 | Chile | Direccion Meteorologica de Chile |
| GMI ma | Mariana Islands | 13.386° North | 144.656° East | 0 | Guam | University of Guam/Marine Laboratory |
| HBA pma | Halley Station, Antarctica | 75.55° South | 25.63° West | 30 | United Kingdom | British Antarctic Survey |
| HPB m | Hohenpeissenberg | 47.8011° North | 11.0245° East | 985 | Germany | Deutscher Wetterdienst |
| HUN | Hegyhatsal | 46.9559° North | 16.6521° East | 248 | Hungary | Institute for Nuclear Research, Hungarian Academy of Sciences |
| ICE pma | Storhofdi, Vestmannaeyjar | 63.3998° North | 20.2884° West | 118 | Iceland | Icelandic Meteorological Office |
| IZO mma | Izana, Tenerife, Canary Islands | 28.309° North | 16.499° West | 2372.9 | Spain | Izana Observatory/Meteorological State Agency of Spain |
| KEY ma | Key Biscayne, Florida | 25.6654° North | 80.158° West | 1 | United States | NOAA Atlantic Oceanographic and Meteorological Laboratory |
| KUM ma | Cape Kumukahi, Hawaii | 19.5608° North | 154.8883° West | 8 | United States | NOAA Global Monitoring Laboratory |
| LLN m | Lulin | 23.47° North | 120.87° East | 2862 | Taiwan | Lulin Atmospheric Background Station |
| LMP ma | Lampedusa | 35.5181° North | 12.6322° East | 45 | Italy | Ente per le Nuove tecnologie, l'Energia e l'Ambiente |
| MEX m | High Altitude Global Climate Observation Center | 18.9841° North | 97.311° West | 4464 | Mexico | Sistema Internacional de Monitoreo Ambiental |
| MHD ma | Mace Head, County Galway | 53.326° North | 9.899° West | 5 | Ireland | National University of Ireland, Galway |
| MID ma | Sand Island, Midway | 28.2186° North | 177.3678° West | 4.6 | United States | U.S. Fish and Wildlife Service |
| MLO mma | Mauna Loa, Hawaii | 19.5362° North | 155.5763° West | 3397 | United States | NOAA Global Monitoring Laboratory |
| NAT ma | Farol De Mae Luiza Lighthouse | 5.7952° South | 35.1853° West | 50 | Brazil | Instituto de Pesquisas Energéticas e Nucleares, Il Centrode Química e Meio Ambiente, Divisao de Quimica Ambiental |
| NMB | Gobabeb | 23.58° South | 15.03° East | 456 | Namibia | Gobabeb Training and Research Center |
| NWR m | Niwot Ridge, Colorado | 40.0531° North | 105.5864° West | 3523 | United States | University of Colorado/INSTAAR |
| OXK m | Ochsenkopf | 50.0301° North | 11.8082° East | 1022 | Germany | Max Planck Institute for Biogeochemistry |
| PAL pm | Pallas-Sammaltunturi, GAW Station | 67.9733° North | 24.1157° East | 565 | Finland | Finnish Meteorological Institute |
| PSA pma | Palmer Station, Antarctica | 64.7742° South | 64.0527° West | 10 | United States | National Science Foundation |
| RPB ma | Ragged Point | 13.165° North | 59.432° West | 15 | Barbados | Private Party |
| SDZ | Shangdianzi | 40.65° North | 117.117° East | 293 | China | Chinese Academy of Meteorological Sciences (CAMS) and Beijing Meteorological Bureau (BMB), China Meteorological Administration (CMA) |
| SEY ma | Mahe Island | 4.6824° South | 55.5325° East | 2 | Seychelles | Seychelles Bureau of Standards |
| SGP | Southern Great Plains, Oklahoma | 36.607° North | 97.489° West | 314 | United States | Lawrence Berkeley National Laboratory |
| SHM ma | Shemya Island, Alaska | 52.7112° North | 174.126° East | 23 | United States | Chugach McKinley |
| SMO ma | Tutuila | 14.2474° South | 170.5644° West | 42 | American Samoa | NOAA Global Monitoring Laboratory |
| SPO pm | South Pole, Antarctica | 89.98° South | 24.8° West | 2810 | United States | National Science Foundation |
| SUM pm | Summit | 72.5962° North | 38.422° West | 3209.54 | Greenland | National Science Foundation Office of Polar Programs |
| SYO pma | Syowa Station, Antarctica | 69.0125° South | 39.59° East | 14 | Japan | National Institute of Polar Research |
| TAP ma | Tae-ahn Peninsula | 36.7376° North | 126.1328° East | 16 | Republic of Korea | Korea Centre for Atmospheric Environment Research Scientific Aviation, Inc, |
| THD ma | Trinidad Head, California | 41.0541° North | 124.151° West | 107 | United States | NOAA Global Monitoring Laboratory, AGAGE, Scripps Institution of Oceanography, Humboldt State University Marine Laboratory |
| TIK pma | Hydrometeorological Observatory of Tiksi | 71.5965° North | 128.8887° East | 19 | Russia | |
| TPI ma | Taiping Island | 10.3786° North | 114.3711° East | 4 | Taiwan | |
| USH ma | Ushuaia | 54.8484° South | 68.3106° West | 12 | Argentina | Servicio Meteorologico Nacional |
| UTA | Wendover, Utah | 39.9018° North | 113.7181° West | 1327 | United States | Beth Anderson/ NWS Cooperative Observer |
| UUM | Ulaan Uul | 44.4516° North | 111.0956° East | 1007 | Mongolia | Mongolian Hydrometeorological Research Institute |
| WIS | Weizmann Institute of Science at the Arava Institute, Ketura | 29.9646° North | 35.0605° East | 151 | Israel | Weizmann Institute of Science and Arava Institute for Environmental Studies |
| WLG m | Mt. Waliguan | 36.2879° North | 100.8964° East | 3810 | Peoples Republic of China | Chinese Academy of Meteorological Sciences (CAMS) and Qinghai Meteorological Bureau (QMB), China Meteorological Administration (CMA) |
| ZEP pmam | Ny-Alesund, Svalbard | 78.9067° North | 11.8883° East | 474 | Norway and Sweden | Zeppelin Station/University of Stockholm Meteorological Institute |

**Appendix B:  Additional results**

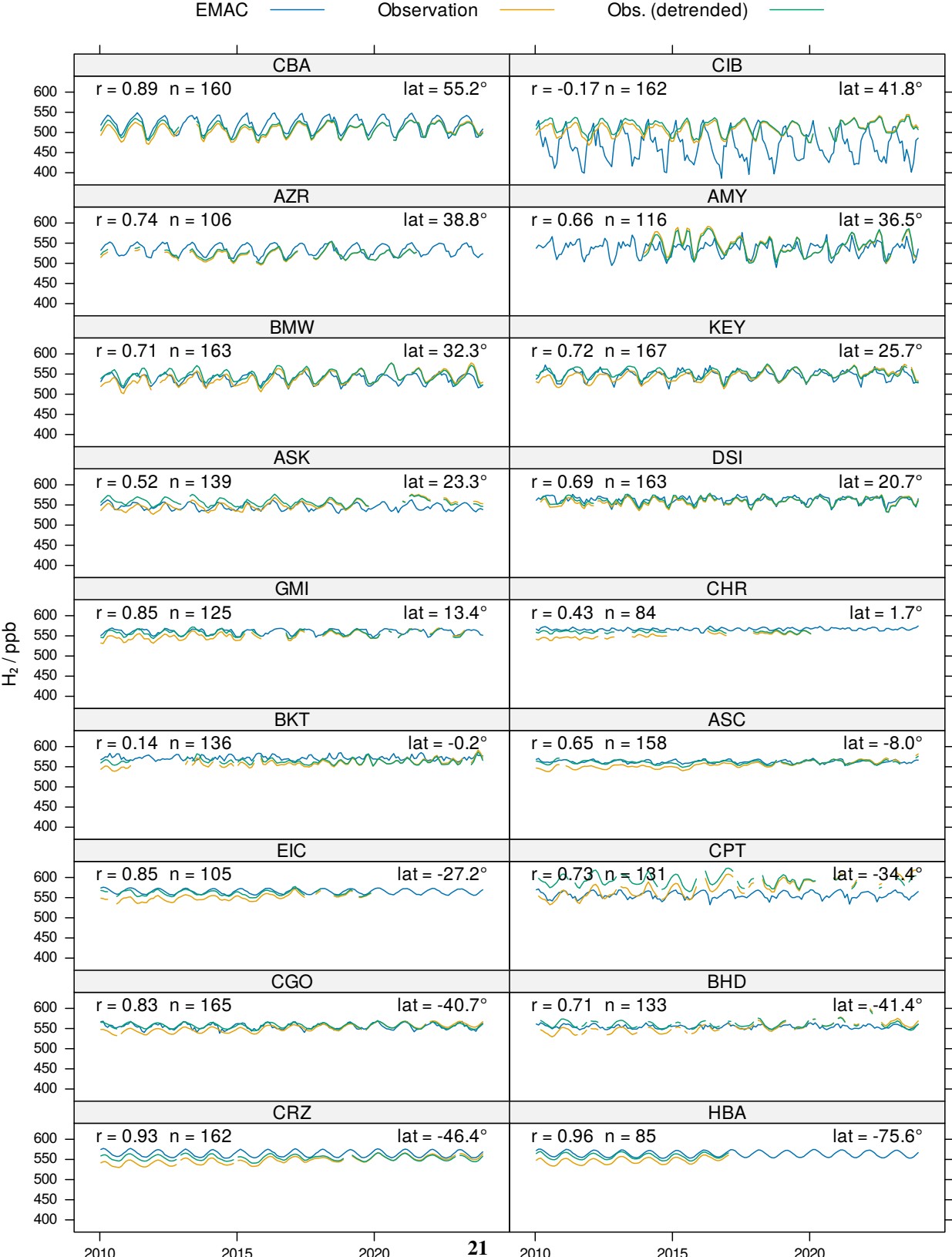

**Figure B1.** Time series comparison of observational and EMAC model data for H$_2$. Latitudes are denoted by lat. [Other stations part 1]

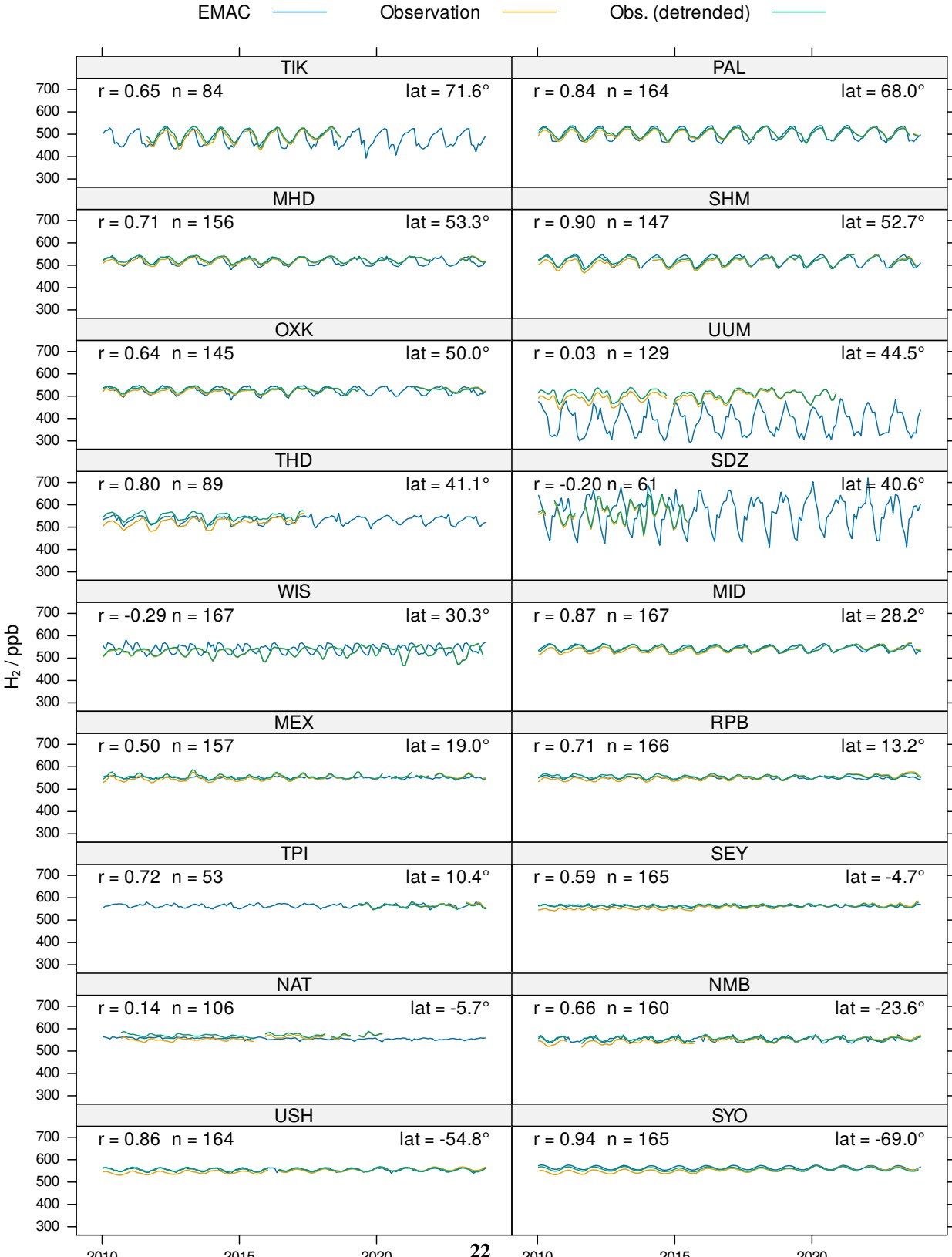

**Figure B2.** Time series comparison of observational and EMAC model data for $H_2$. Latitudes are denoted by lat. [Other stations part 2]

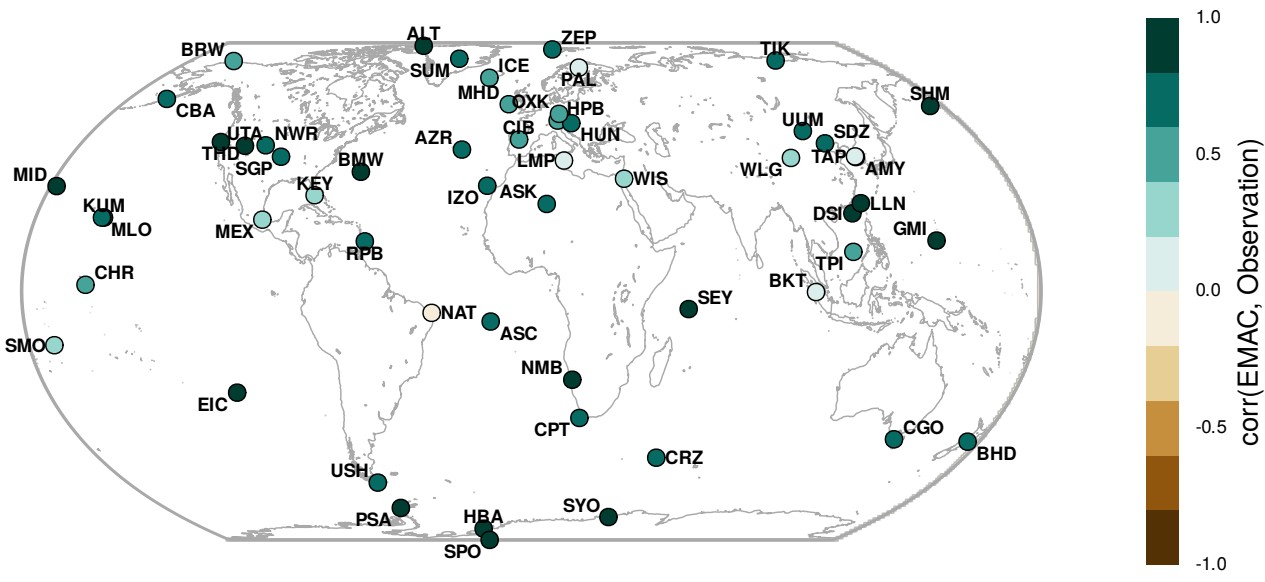

**Figure B3.** Pearson correlation coefficient between EMAC CH$_4$ mixing ratios and observational data. Model data is compared with detrended observational data for the years 2010–2023 (inclusive) to perform these calculations. For a more extensive comparison see Zimmermann et al. (2020).

**Table B1.** Comparison of mean model and observational $H_2$ mixing ratios. $\Delta$ = Model $-$ Observed, while $r$ denotes the Pearson correlation coefficient. In the case of the BKT station, the EMAC value of one grid cell to the west of the station is used as it is considered more representative.

| Station | Longitude | Latitude | # values | $H_2$ EMAC (ppb) | $H_2$ Observed (ppb) | $\Delta$ (ppb) | $r$ |
|---|---|---|---|---|---|---|---|
| ALT pma | -62.5 | 82.5 | 168 | 504. | 501. | 3.19 | 0.93 |
| ZEP pmam | 11.9 | 78.9 | 168 | 515. | 513. | 1.87 | 0.94 |
| SUM pm | -38.4 | 72.6 | 168 | 522. | 526. | -4.00 | 0.81 |
| TIK pma | 129. | 71.6 | 84 | 479. | 499. | -19.8 | 0.65 |
| BRW pma | -157. | 71.3 | 168 | 502. | 507. | -4.82 | 0.81 |
| PAL pm | 24.1 | 68.0 | 164 | 503. | 506. | -3.01 | 0.84 |
| ICE pma | -20.3 | 63.4 | 168 | 518. | 522. | -3.95 | 0.86 |
| CBA ma | -163. | 55.2 | 160 | 520. | 510. | 9.96 | 0.89 |
| MHD ma | -9.90 | 53.3 | 156 | 517. | 525. | -8.03 | 0.71 |
| SHM ma | 174. | 52.7 | 147 | 521. | 518. | 2.89 | 0.90 |
| OXK m | 11.8 | 50.0 | 145 | 525. | 528. | -3.19 | 0.64 |
| HPB m | 11.0 | 47.8 | 168 | 526. | 527. | -1.10 | 0.56 |
| HUN | 16.7 | 47.0 | 168 | 513. | 523. | -9.78 | 0.39 |
| UUM | 111. | 44.5 | 129 | 388. | 508. | -121. | 0.034 |
| CIB | -4.93 | 41.8 | 162 | 467. | 514. | -46.9 | -0.17 |
| THD ma | -124. | 41.1 | 89 | 532. | 547. | -15.0 | 0.80 |
| SDZ | 117. | 40.6 | 61 | 566. | 569. | -2.98 | -0.20 |
| NWR m | -106. | 40.1 | 168 | 547. | 536. | 11.2 | 0.59 |
| UTA | -114. | 39.9 | 168 | 527. | 514. | 12.8 | 0.82 |
| AZR ma | -27.4 | 38.8 | 106 | 533. | 523. | 10.4 | 0.74 |
| TAP ma | 126. | 36.7 | 168 | 535. | 547. | -12.7 | 0.62 |
| SGP | -97.5 | 36.6 | 168 | 507. | 519. | -12.2 | 0.77 |
| AMY ma | 126. | 36.5 | 116 | 536. | 543. | -7.17 | 0.66 |
| WLG m | 101. | 36.3 | 168 | 527. | 529. | -1.29 | 0.56 |
| LMP ma | 12.6 | 35.5 | 168 | 517. | 532. | -14.5 | 0.046 |
| BMW ma | -64.9 | 32.3 | 163 | 539. | 546. | -7.09 | 0.71 |
| WIS | 35.0 | 30.3 | 167 | 544. | 530. | 13.8 | -0.29 |
| IZO mma | -16.5 | 28.3 | 168 | 545. | 551. | -6.18 | 0.61 |
| MID ma | -177. | 28.2 | 167 | 544. | 543. | 0.830 | 0.87 |
| KEY ma | -80.2 | 25.7 | 167 | 548. | 552. | -3.73 | 0.72 |
| LLN m | 121. | 23.5 | 168 | 563. | 565. | -1.83 | 0.63 |
| ASK m | 5.63 | 23.3 | 139 | 545. | 558. | -13.1 | 0.52 |
| DSI ma | 117. | 20.7 | 163 | 563. | 560. | 2.75 | 0.69 |
| KUM ma | -155. | 19.6 | 168 | 544. | 544. | 0.133 | 0.75 |
| MLO mma | -156. | 19.5 | 168 | 558. | 548. | 10.4 | 0.70 |
| MEX m | -97.3 | 19.0 | 157 | 551. | 556. | -5.06 | 0.50 |
| GMI ma | 145. | 13.4 | 125 | 560. | 557. | 3.36 | 0.85 |
| RPB ma | -59.4 | 13.2 | 166 | 550. | 557. | -6.91 | 0.71 |
| TPI ma | 114. | 10.4 | 53 | 565. | 563. | 1.58 | 0.72 |
| CHR ma | -157. | 1.70 | 84 | 567. | 560. | 7.12 | 0.43 |
| BKT mma | 100. | -0.202 | 136 | 572. | 563. | 8.29 | 0.14 |
| SEY ma | 55.5 | -4.68 | 165 | 563. | 564. | -1.10 | 0.59 |
| NAT ma | -35.2 | -5.68 | 106 | 554. | 570. | -16.9 | 0.14 |
| ASC ma | -14.4 | -7.97 | 158 | 562. | 561. | 1.26 | 0.65 |
| SMO ma | -171. | -14.2 | 168 | 567. | 561. | 5.54 | 0.65 |
| NMB | 15.0 | -23.6 | 160 | 554. | 554. | 0.158 | 0.66 |
| EIC ma | -109. | -27.2 | 105 | 565. | 561. | 4.23 | 0.85 |
| CPT ma | 18.5 | -34.4 | 131 | 557. | 593. | -35.8 | 0.73 |
| CGO ma | 145. | -40.7 | 165 | 555. | 557. | -1.56 | 0.83 |
| BHD ma | 175. | -41.4 | 133 | 554. | 561. | -6.38 | 0.71 |
| CRZ ma | 51.8 | -46.4 | 162 | 564. | 554. | 10.3 | 0.93 |
| USH ma | -68.3 | -54.8 | 164 | 554. | 556. | -1.20 | 0.86 |
| PSA pma | -64.1 | -64.8 | 168 | 564. | 557. | 6.43 | 0.95 |
| SYO pma | 39.6 | -69.0 | 165 | 564. | 558. | 6.05 | 0.94 |
| HBA pma | -26.2 | -75.6 | 85 | 564. | 558. | 5.79 | 0.96 |
| SPO pm | -24.8 | -89.5 | 168 | 564. | 558. | 5.81 | 0.96 |

*Author contributions.* JL managed the project with contributions from AP. BS led the delivery of model simulations with contributions from AP, KK, SG, CB and NS. KK led the delivery of soil sink modelling with contributions from BS. CB led the collation of observational data and its quality control with contributions from BS. SG led the delivery of chemical tagging with contributions from BS. NS wrote the manuscript with contributions from all co-authors. All authors met to discuss the results and contributed to the writing and editing of the manuscript.

*Competing interests.* We declare that two of the co-authors hold an editorial board position with Geoscientific Model Development. The authors have no other competing interests to declare.

## Acknowledgements

N. S. acknowledges funding support from the CSIRO International Hydrogen Research Fellowship scheme and the Faculty of Engineering and Information Technology at UTS to undertake a sabbatical at the Max Planck Institute for Chemistry in Mainz. All simulations were performed with the Levante High Performance Computing System (https://www.dkrz.de/en/systems/hpc/hlre-4-levante) hosted by Deutsches Klimarechenzentrum GmbH.

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
