# Peer review of "Global atmospheric hydrogen chemistry and source-sink budget equilibrium simulation with the EMAC v2.55 model"

_EGUsphere, 2025_

## Author Comment (AC1)

We thank both reviewers and the chief editor for their constructive comments on this article. We provide a point-by-point response to their comments below. In bold are the comments of the reviewers, followed by our replies.

**Reviewer # 1**

**Line 6: remove comma after scheme**

No change. The sentence reads correctly with the comma.

**Line 14: 'by realistically simulating OH in the atmosphere' – after reading the paper, the support offered for this statement is a comparison of the tropospheric chemical lifetime of methane in EMAC (8.9 years) with an 'observational estimate' of 9.1 years from Holmes et al. (2013). Could the authors confirm that these two values are directly comparable (i.e. represent the same loss mechanisms)? The 9.1 yr value from Holmes et al. appears to come from Prather et al. 2012 – and is an estimate of the total atmospheric lifetime for methane (i.e. includes losses to the stratosphere and soil). The observational value for methane lifetime against OH based on methyl chloroform data is quoted as 11.2 years in Holmes et al. (2013) (and is also taken from Prather et al. 2012).**

The abstract has been modified. The use of methane lifetime as an estimate for the OH radical is explicitly mentioned. In section 4.1, we have extended the discussion on $CH_4$ lifetime and explicitly explain the dominance of OH as methane sink as the reason why methane lifetime is regarded as a proxy for the global oxidising capacity of the atmosphere (Lelieveld et al., 2016; Saunois et al., 2025).

We also thank the reviewer for pointing us to Prather et al. (2012). The reference to Holmes et al. (2013) has been removed and replaced by Prather et al. (2012) for the total observational constrained methane lifetime of $9.1 \pm 0.9$ years. We also directly refer to the supporting information of Prather et al. (2012) for the tropospheric chemical lifetime: estimate of lifetime for reaction with tropospheric OH from methyl chloroform observations, $11.2 \pm 1.3$ years and reaction with tropospheric chlorine of $200 \pm 100$ years.

**Line 49: 'We find that correctly representing the oxidising capacity of the atmosphere is critical for predicting H2 mixing ratios and their spatio-temporal variability'. I'm not sure that this has been done – you have captured observations of H2 mixing ratios using the model's OH and representations of the other sources/sinks, but you have not shown how sensitive H2 is to differences in OH? Spatially, and seasonally, H2 will likely be very sensitive to the soil sink – which is estimated to be significantly larger than the OH sink, as well as photochemical production of H2.**

We have modified the text accordingly and explicitly present the global budget of the soil sink and atmospheric production of hydrogen, which are in fact quite comparable in our simulation.

**Line 64: Just a note that diurnal variability is seen in longer-lived gases at the surface due to diurnal variations in boundary layer height and the resulting changes**

to mixing (e.g. H2 measurements form Weybourne - Forster et al, Tellus B, 2012).

Thanks for making the authors aware of previous work undertaken by Forster et al. We find that the diurnal variations in $H_2$ measurement presented by Forster in Figure 5 are captured well within the uncertainty bounds of the measurements. Based on this finding, and due to computational constraints, we believe a daily time-step is appropriate for our model analysis.

**Line 66: I think there should probably be an acknowledgement here that in 2020, methane was (and still is) not in steady-state in the atmosphere. Therefore a simulation run to steady-state using repeating 2020 emissions is not fully representative of 2020 atmospheric conditions.**

We have changed the text accordingly.

**Line 71: 'swamps or wetlands': these 2 words are not interchangeable - swamps are a type of wetland. Maybe change to 'wetlands other than bogs'?**

We thank the reviewer for noticing this. Obviously, the category swamps contains all wetlands which are not bogs. We renamed accordingly.

**Line 74: Change 'swamps' to 'wetlands' here (see comment above)?**

We have made the requested change on line 74.

**Line 86: Do the RETRO/GFED databases include estimates for H2 emissions? Usually, H2 emissions are obtained by using H2:CO emission factors to scale CO emission inventories – has the same been done here, and if so which emission factors are used?**

Yes, we can confirm that both the RETRO and GFED databases provide direct estimates of $H_2$ emissions without relying upon an assumed $H_2/CO$ emissions ratio. We added: 'Both the RETRO and GFED databases provide direct estimates of $H_2$ emissions without relying upon an assumed $H_2/CO$ emissions ratio.'

**Line 90: Could the authors clarify if the emissions for non-GHG species are transient or for a single year/climatology?**

We added at the first paragraph of section 2.1: 'Therefore, emissions are based on the year 2020 or the closest year prior to 2020 and are repeated for each year, removing any interannual variability.'

**Line 100, Section 2.2: It would be nice to see some analysis of the soil deposition velocities calculated within the model, including geographical variation etc. Have the authors compared model results to any deposition velocity measurements?**

As there are only a few available measurements of deposition velocity, we validated the resulting $H_2$ mixing ratios instead. Paulot et al. (2021) have previously validated the deposition velocity parametrisation used here by comparing it with deposition measurements taken at a small number of stations.

**Line 129: We are told the scaling factor 'A' is adjusted to obtain a global mean**

deposition velocity of 0.033 following Yashiro et al. (2011). What was the reason for choosing this value to scale to? The Yashiro study tuned their global soil uptake to optimise the model agreement with observations (by adjusting their inactive layer thickness, delta). Also, what value of 'A' was required for the adjustment – i.e. was a large or small adjustment required?

Initially, the deposition velocity was calibrated by adjusting the parameter $A$ (representing bacterial activity) to provide global dry deposition estimates that agreed well with existing estimates by (Paulot et al., 2021) and Ehhalt and Rohrer (2011). To improve the agreement with $H_2$ station data, the parameter $A$ was further adjusted to obtain a small reduction in the globally averaged deposition velocity by about 5% to 0.033 cm$s^{-1}$, consistent with Yashiro et al. (2011). The resulting value of the resolution dependent parameter $A$ is 10.9 using ERA5 input data with 0.25 degree grid spacing.

**Line 146: Does the soil uptake calculated vary with time during the simulation, or is the ERA5 data averaged over a set period?**

The soil uptake varies with time according to the ERA5 input data which is synchronised with the ERA5 nudging data.

**Figure 1: Much of the globe is a similar shade of green. You can see the IHG in the modelled data – but all observed data looks the same/very similar on this colour scale. Would it be possible to adjust the colour scale so that it is easier to discern differences in H2 between observational sites and see a bit more structure?**

Agree – change made.

**Figure 2: It would be helpful to have some further info on the station rather than just the station code in this figure, e.g. latitude/longitude. I think it would also help the reader if the plots were ordered by latitude going from north-south rather than alphabetical order. This would make it easier to discern latitudinal differences in seasonality and how the model is performing.**

Agree. We have ordered the plots by latitude as requested.

**Line 225-230: As mentioned above, it's not clear to me that the 8.9 and 9.1 year lifetimes quoted are comparable. The EMAC methane lifetime of 8.9 years is within the range of the other models referred to by Yang et al. 2024 that Yang argues underestimate the methane lifetime (8.3 to 9.5 years).**

We thank the reviewer for this comment. The last paragraph of section 4.1 has been modified and we have consequently based our lifetime calculation on total atmospheric burden for methane and hydrogen. Therefore our estimated $CH_4$ lifetimes are directly comparable to Prather et al. (2012). We have also changed numbers in Table 1 accordingly and noted that all calculations are performed with respect to global burden.

**Line 247: 'The small long-term trend in H2 captured by the model' – I thought this was a steady-state simulation with methane and hydrogen emissions held constant (I'm not sure if non GHG emissions are transient)? We know CH4 (which is a source of H2) is increasing in the atmosphere, and this will not be captured in a steady-state**

simulation.

Agree. This sentence has been removed.

**Line 254-5 (and elsewhere wherever bottom-up is mentioned): The authors refer to bottom-up and top-down estimates for the global soil sink (55-60 Tg/yr and 85-88 Tg/yr respectively). I think it is worth pointing out that many of the 'bottom-up' studies have scaled the global soil sink uniformly to capture H2 atmospheric observations (which makes them top-down global soil sink estimates from my perspective - although the geographical variability will be bottom-up). The only truly bottom-up global soil sink estimate I am aware of that did not scale the soil sink to match total sources is Sanderson et al. (2003). The 'top-down' studies that are referenced are HD and inversion studies in which the larger soil sink was balanced by larger estimates of the photochemical production of H2 to close the H2 and HD budgets. The impact of including a geological H2 source in these studies is not clear without further information on the spatial/temporal variability and isotopic composition of the geological source. The way this sentence is phrased – 'the unaccounted for source almost bridges the gap between bottom-up and top-down estimates' implies that the larger soil sink estimates are consistent with the existence of a geological source, which I don't think can be assumed.**

We agree that, as it has been formulated, the geological hydrogen is highly speculative and that there might be a misinterpretation of bottom-up versus top-down estimates. We still think that a potential geological source, if it exists, would be beneficial for the top-down inversion studies. There would be no (lesser) need for an up-scaling of photochemical production, as mentioned by the reviewer. Nevertheless, since this $H_2$ source and its isotopic composition has not been confirmed, we removed the sentence and just mention this potentially unaccounted source term. We have rewritten: 'Ellis and Gelman (2024) estimate that global geological $H_2$ resources cause an additional global flux of 24 Tg $yr^{-1}$ from the subsurface to the atmosphere. This is speculative and would add unaccounted hydrogen emissions almost of the strength of the current non-photochemical sources. Current knowledge concerning the budget of atmospheric $H_2$ does not exclude the existence of a large geological $H_2$ reservoir, and further emphasises the importance of the dry deposition for the global atmospheric $H_2$ budget.'

**Line 280-1: 'the CH4 chemical lifetime in excellent agreement with observational estimates' – see my comments above about the CH4 lifetime.**

We thank the reviewer for the very useful hint. We have improved the discussion on methane lifetime in section 4.1. We consequently refer the reader to the original paper of Prather et al. (2012) and have extended the comparison regarding tropospheric chemical lifetime.

**Line 281: 'We conclude that correctly simulating the oxidising capacity is a key requirement for high accuracy simulation of H2' – I'm not sure why OH is emphasised here above other sources/sinks in the H2 budget.**

We agree with the comments above about oxidising capacity that was not systematically explored and the major impact of the soil sink of $H_2$. But we believe that the good results concerning the meridional gradient of $H_2$ are supporting that the simulated interhemispheric differences in OH are adequate. The text has been changed accordingly. We have replaced the sentence: 'The $H_2$

soil sink, based on a two-layer soil model (Yonemura et al., 2000; Ehhalt and Rohrer, 2013; Paulot et al., 2021), in combination with monthly ERA5 reanalysis data for soil related parameters has been successfully used by the EMAC model. We conclude that atmosphere chemistry models with such features, capturing the most dominant terms of the atmospheric $H_2$ budget, should be able to generally simulate station observations of atmospheric hydrogen. This gives confidence that scenario simulations on future hydrogen economy provide reliable estimates of the atmospheric impact.'

**Reviewer # 2**

**Title - 'long-term source sink budget simulation' is slightly misleading, given the input emissions for hydrogen are fixed to the year 2020 (but taken from the year 2000 in the RETRO dataset). I would suggest changing this title to reflect the actual simulation, or adding a comment discussing the trend in the deposition sink over the period.**

We agree that the title might be slightly misleading. It has been changed to equilibrium simulation. We think that the wording long-term is still justified, since we are using flux boundary condition for $H_2$ and $CH_4$ emissions. The performance and long-term stability of such a setup for two closely coupled chemical species in the atmosphere is the topic of this paper. This is of special importance for applications of the EMAC model concerning scenarios regarding a future hydrogen economy.

**Line 4: 'Long-term global simulations were performed, with a horizontal resolution of 1.9 degrees. The results of this simulation are compared with'. Same comment here regarding 'long-term'.**

Please see, the comment above. We changed the text to: 'Long-term global equilibrium simulations'.

**Line 6: Clarify that the soil sink is not the only sink present (as it currently reads), but that there is also a chemical sink for hydrogen in the model.**

Agree. We changed the text to: 'We introduced $H_2$ sources and sinks, that latter which was inclusive of a soil uptake scheme that accounted for bacterial consumption'.

**Line 38: State that all three species listed are radiatively active, not just stratospheric water vapour.**

We have changed the text accordingly: '... increases in radiatively active species ...'

**Line 39: Change 'can' to 'may'. The impact of hydrogen on aerosols is highly uncertain currently. In the next line 'Arising from these observations' should be changed to 'Arising from these modelled results' or similar.**
**Line 198: 'from' Zimmerman et al.**

Agree – changes made.

**Line 10: Here, and later in the paper, discussion of 'direct anthropogenic perturbation' could be expanded. Terminology such as 'unpolluted air masses' - see Derwent et al. 2023 (doi: 10.1016/j.atmosenv.2023.120029) may read better. I think the paper could benefit from a discussion (or an additional figure of the emissions input**

such that sites near high emission regions can be clearly identified) of how it was determined which sites fit this criteria for being background/ undisturbed stations. This discussion regarding sites near high emission regions could also benefit from a statement on the long lifetime of hydrogen.

We thank the reviewer for the hint. Unpolluted has been added, and the term direct anthropogenic perturbation has been omitted. Air masses can be polluted without anthropogenic influences, e.g. naturally occurring wildfires etc.

**Line 15: Remove 'realistically'.**

This text has been changed according to the comment from Reviewer 1 at line 14.

**Line 58: Link the sentences on spin-up to steady-state. In the intermediate sentence here, expand upon the issues which arise from prescribed boundary conditions. This is key to the benefits of this model simulation and should be highlighted.**

We thank the reviewer for this comment. See our response to the paper title. Here we added additional information on the strong chemical coupling of $H_2$ and $CH_4$: '$H_2$ and $CH_4$ are chemically coupled and have nearly the same chemical lifetime (Table 1). Both compete for the hydroxyl radical OH as a chemical sink term, with OH being by far the dominant sink for atmospheric methane (Saunois et al., 2025, see section 4.1 below). Furthermore, is the atmospheric oxidation of $CH_4$ which is the largest source for $H_2$ (Ehhalt and Rohrer, 2009). To adequately simulate such a coupled system, the EMAC model uses flux boundary conditions for sources and sinks of both species.'

**Line 67: The COVID-19 pandemic in 2020 could first be mentioned here to clarify that emissions are estimates without lockdowns. It may also be useful to clarify here that although interannual variability for emissions is removed, there is still meteorological variability (and potentially other sources of variability?).**

We have to apologise for a typographical error. The emissions of anthropogenic sources CAMS-GLOB-ANTv4.2 and CAMS-GLOB-AIRv1.1 are used with COVID-19 reductions. That's why Reifenberg et al. (2022) has been cited. The reduction in concentration of the hydroxyl radical OH is below 4% for most of the atmosphere with the exception of the uppermost troposphere and tropopause region, which is due to reduced flight activity. The small impact on OH is foremost confined to northern hemisphere mid latitudes. We added at line 91: '... and the emissions are estimated with reduction due to lockdowns during the COVID-19 pandemic (Reifenberg et al., 2022). The reduction in concentration of the hydroxyl radical OH is below 4% for most of the atmosphere with the exception of the uppermost troposphere and tropopause region, which is due to reduced flight activity. The small impact on OH is foremost confined to northern hemisphere mid latitudes.'

**Line 70: Is this methane simulation published? If so, please cite this publication here. If not, a justification of this statement and further discussion on the methane representation is needed.**

We refer to the publication of Zimmermann et al. (2020), which describes a transient simulation with methane emissions. The text has been changed to clarify this: "For the long-lived tracer $CH_4$, the a posteriori emissions and the best combination of the rising-methane scenario of Zimmermann et al. (2020) have been applied. In this work, Zimmermann et al. (2020) show that the EMAC model has been efficient in simulating interactive $CH_4$ mixing ratios over the last

two decades. Therein, the model results compare quite well with NOAA and AGAGE stations and measurements from CARIBIC (Civil Aircraft for the Regular Investigation of the atmosphere Based on an Instrument Container) flight observations (Brenninkmeijer et al., 2007)."

**Line 82: Based on these slightly older emissions, a more detailed discussion on the comparison between this dataset and those used in more recent literature would be beneficial. There is mention later on of Paulot et al. (2021), but there is also the slightly updated values in Paulot et al. (2024) (https://doi.org/10.5194/acp-24-4217-2024). This paper, alongside results presented in Sand et al. (2023), could also be added to Table 2.**

Agree. The requested data from Sand et al. (2023) and Paulot et al. (2024) has been added to Table 2.

**Line 85: There have been changes in hydrogen emissions in the last few decades (see Paulot et al. 2021 and 2024). Following on from above, a more direct comparison between the dataset used in this paper and in the literature is needed.**

We have consulted both papers by Paulot et al. (2021, 2024) as recommended by the reviewer. In both cases, we respectfully disagree that the suggested papers change our claim regarding the global H$_2$ budget.

**Line 90: Include a sentence on the aerosol set-up used in the model.**

We omitted: 'The ONEMIS submodel (Kerkweg et al., 2006) calculates natural emission fluxes of sea salt (Guelle et al., 2001) and dust (Klingmüller et al., 2018; Astitha et al., 2012).' It is correct that the ONEMIS submodel provides this data in the simulation, but it is not used. It is replaced by: 'The simulation uses a climatology of the aerosol wet surface density to calculate heterogeneous reactions. It is based on the CMIP5 (Climate Model Intercomparison Project Phase 5) emissions climatology for the years 1996–2005 low S scenario (Righi et al., 2013). The aerosol distribution for radiative forcing calculation is the Tanre climatology (Jöckel et al., 2006)'.

**Line 139: Rephrase the threshold moisture content to clarify that there is no uptake in the top, dry layer in the model.**

We clarified the sentence; "The soil water content in the top, dry layer (i.e. $\theta_{wI}$) is assumed to be the threshold moisture content below which the bacterial activity vanishes i.e. there is no H$_2$ uptake in this layer, and is given by: ...".

**Figure 2: The line colours on this plot need to be more distinctive. A suggestion here is to split the stations into northern and southern hemisphere and to add lines to indicate January 1st of each cycle. This plot also demonstrates the changes in the observations in the period leading up to 2020. There has been an increase in background hydrogen concentrations in that decade, evident in the observations shown in sites such as SPO and PSA in Figure 2. The model does not capture this increase, due to emissions being fixed at one year. This is interesting, as it shows that this may not be related to any changes in the hydrogen soil sink over this period (I am assuming these interannual changes are simulated in this run, but this should be clarified).**

We rearranged Figure 2 and Figures B1 and B2 so that the stations now appear from north to

south as requested. Furthermore, we clarified that for the equilibrium simulation for the hydrogen soil sink, data for the year 2020 is repeated. The observed slight increase in atmospheric hydrogen, especially obvious in the Antarctic observations, coincides with the increase in atmospheric methane since 2010 after the hiatus during the previous decade. This is added to : 'Especially the Antarctic stations show an upward trend for atmospheric hydrogen. The model with its emissions and soil sink data repeating the year 2020, cannot capture this feature. This trend coincides with further increasing atmospheric methane concentrations after 2010 following the hiatus of the previous decade (Lan et al., 2024). Atmospheric $CH_4$ is the largest source for $H_2$ (see Table 2 in Ehhalt and Rohrer, 2009). To investigate this further in the future, a model simulation with flux boundary conditions for $H_2$ and $CH_4$ in transient mode is needed'.

**Line 191: Or related to uncertainty in the soil sink? Broadly the soil sink is considered to be the most uncertain part of the hydrogen budget, however in Table 1, the uncertainty is very low. Comment on how this uncertainty was calculated and why it differs from other budgets.**

We thank the reviewer for this comment. As described in the caption of Table 1, the uncertainty in Table 1 is calculated as the standard deviation of multi-year annual global means. Since the boundary conditions of emissions and deposition fluxes are repeated (i.e. an equilibrium simulation), the standard deviations are small. In our simulation, the uncertainties represent just a statistical measure, and do not represent those caused for example by limitations of the soil model or other parametrisations or ERA5 data etc. We have added that the uncertainty in the soil sink is still large and that large land masses have no observational stations: 'The coverage of many continental land masses by the observational stations is sparse. For South America, Australia, Africa, Central Asia, Siberia and India there are almost no measurements available. This is a problem, especially in validating the soil sink, which can be considered the most uncertain part of the hydrogen budget (Paulot et al., 2021).'

**Line 218: Can a comparison be made to more up-to-date literature than Yashiro (2011)?**

We confirm that more recent literature is available for comparison e.g. Pieterse et al. (2013); however, the paper by Yashiro et al. (2011) represented the most useful basis for comparison with our study since it provided multi-year time series model/data comparisons for a large number of observational stations with the same fitting metric we employed. In contrast, Pieterse et al. (2013) presents model/data time series comparisons for only two years for a much smaller number of sites and with a different fitting metric.


This is correct. There is no trend in the model data. The sentence has been removed. The misleading sentence is an error which occurred during the final editing of the paper. There is a weak trend in atmospheric $H_2$ in the last 15 years, which might be caused by the further increase of atmospheric methane concentration after the hiatus, but not in the model.

**Line 255: In the previous paragraph, it was stated that this simulation agreed well with bottom-up estimates. If the additional source of natural hydrogen is present, a large shift elsewhere in the budget is needed, however the very small uncertainties presented in the hydrogen budget for this work do not leave room for any shifts.**

We thank the reviewer for this comment. Please see the comment of reviewer 1, lines 254–5, who has stated that for bottom-up estimates, usually the soil sink is scaled to capture $H_2$ atmospheric observations. Whereas, in the case of top-down studies, the estimated soil sink is balanced by a scaling of atmospheric chemical production. Therefore, we have removed the remark concerning bridging the gap between bottom-up and top-down estimates. Because in our simulation the total atmospheric oxidation capacity based on methane lifetime seems to be in line with estimates based on methylchloroform observations, we have to conclude that a higher soil sink is needed to compensate for this unaccounted source. Based on the top-down estimates for the soil-sink this is possible.

**Figure B2: WIS appears to have the same number of observations (167) as the sites included in Figure 2. Is there a reason this is not included in that figure?**

**WIS has 167 observations, whereas Figure 2 includes sites with 168 observations. This was not apparent as the model data only covered 167 months. However, the simulation has now been extended to December 2023, providing 168 months of coverage. All 168 months are now shown in Fig. 2. A discussion of Figure B3 would be beneficial.**

We have added to the caption of Figure B3: 'Model data is compared with detrended observational data for the years 2010-2023 (inclusive) to perform this calculations.' For a more extensive comparison see Zimmermann et al. (2020).

**Chief Editor**

Dear authors,

Regarding compliance with the journal's Code and Data policy in your submitted manuscript, could you please clarify the current situation regarding the publication of the EMAC model? We have now seen many papers submitted to the journal storing the code in private repositories. In the past, authors of papers from the development team claimed that steps were been taking to address this situation and make the EMAC code public and open. What is the current status of this issue? Is the situation that you can not obtain permission to publish the code from former developers?

Dear Juan,

Thank you very much for asking about progress in releasing the MESSy code (including EMAC) under an open source licence. As you might imagine, releasing such a code base, of which some legacy parts date three or more decades back in time, relate to times when nobody in science really thought about source-code licensing, is a major issue. A lot of legal issues need to be tackled to solve this matter. This is in particular difficult, if more than 10 institutions, each of which has its own legal department, are involved. We are on a good path to addressing this matter, but some more legal issues need to be resolved. It is important to understand that such an open source release needs to be "bullet proof" in terms of voidability from third parties. At present, none of our staff who are legally authorised to sign agreements will do that easily without being convinced about the agreement "being safe".

[revised manuscript text omitted]

---

## Author Response (AR2)

We thank the editor for her constructive comments on this article. We provide a point-by-point response to the comments below.

**Editor comments**

**1. Reviewer #1 suggested a comparison between the derived deposition velocities with observations. You indicate that this comparison was previously done in Paulot et al. (2021) and that you are using the same soil sink scheme. However, the input data being used here differs from that in Paulot et al., and hence the performance may differ. I agree with the reviewer here that a comparison with observations and/or Paulot et al. is warranted.**

Indeed, due to differences in the input data, resolution, and time period, differences in the resulting deposition velocity are to be expected. It is not possible to make a direct comparison with the results of Paulot et al. (2021) and related studies, as none of the data presented in that manuscript are publicly available. However, we have now included a map showing our average deposition velocity with the same colour scale as in Figure 2 of Paulot et al. (2021). This map can be found in the supplement, alongside a plot of the zonal mean deposition velocity that allows further comparison with the results presented by Paulot et al. (2021). We added a paragraph to section 2.2 in the main text briefly discussing these figures: namely, that

'The global distribution of the $H_2$ deposition velocity averaged over the 2012 to 2021 period resembles the maps presented by Paulot et al. (2021), but with more pronounced extrema (Figure S1 in the supplement). The zonal mean of the $H_2$ deposition velocity over land also falls within the range of results reported by Paulot et al. (2021). However, the local minimum around 20°N, due to the Sahara, and the maximum around 10°N, where the transition to more humid regions favours soil uptake, are more distinct in this study than in most of those shown by Paulot et al. (2021) (Figure S2 in the supplement). Thanks to extensive observational records of atmospheric hydrogen concentrations at stations around the world, it is possible to carry out detailed validation of hydrogen deposition based on the resulting concentrations. This method, presented in the following sections, is more informative than local deposition analysis because it is less susceptible to significant variations in surface conditions and is based on a much larger database.'

The deposition data used in our simulations and presented in the figures is now available in the Edmond Open Research Data Repository of the Max Planck Society. We now also emphasise that validating the resulting atmospheric concentrations is more informative than analysing the local deposition, since the former is less affected by strong variations in surface conditions and has a much larger observational database.

**2. Reviewer #2 made a comment on the use of the wording "long term" in the title, given that the simulation performed here is a timeslice (albeit with time-varying meteorology). With respect, I do not agree with the statement "We think that the wording long-term is still justified, since we are using flux boundary condition for $H_2$ and $CH_4$ emissions." The emissions and the soil sink used are representative of only a single year and used repeatedly throughout the equilibrium simulation. As a result, the simulation performed for this study will not capture changes in secondary production of hydrogen, for example, as would arise if the simulation was transient and using time-varying emissions. Therefore, I kindly ask that you adjust your title**

**accordingly and that you remove "long term" in the main text based on Reviewer #2's second comment (i.e., about Line 4 of original manuscript).**

We thank the editor for this comment. The title has been changed accordingly: 'Global atmospheric hydrogen chemistry and source-sink budget equilibrium simulation with the EMAC v2.55 model'. The abstract has been modified: 'Extensive global equilibrium simulations were performed with a horizontal resolution of 1.9 degrees. The results of this simulation are compared with observational data from 56 stations'. We think that the formulation extensive is justified, since the simulation, including spin-up time, covers 18 years (2006 - 2023). Actually, if we would here also account for the time to gain improved initial conditions for the atmospheric methane concentration, this time is even longer (34 years, 1990-2023). In the paper the spin-up time (2006-2009) refers to the time that is needed to adjust the hydrogen ocean source according to literature recommendations and the slight scaling of the soil sink (in comparison to Paulot et al. (2021)).

**3. Linked to the request above is a request to add greater clarity around the simulation set up. For example, the text states "the model experiment covers the time period 2006–2023". This is misleading and should be changed to something like "The model experiment is representative of the present-day (Year 2020) and uses meteorology for the years 2006-2023, with the first three years used as spin-up time."**

We follow the suggestion and have changed accordingly: 'The model experiment is representative of the present-day (i.e. the year 2020) and uses meteorology for the years 2006-2023, with the first four years used as spin-up time'. The spin-up time has been corrected. It is four years 2006-2009.

**4. It is still unclear what criteria were used to determine which sites fall into which categories (i.e., well mixed polluted/unpolluted) based on Reviewer #2's request – may I please ask that you expand on this?**

We thank the editor for this comment. The stations are now characterised as marine, polar, mountain, or none of these categories. Generally, the nature of the measured air masses is discussed along the line of the sites characterisations. It is clarified that the station network is predominantly measuring remote or free tropospheric air masses. Furthermore, sites impacted with high air pollution are more clearly identified. The terms well-mixed and unpolluted are completely omitted from the manuscript. It's briefly discussed that the very low number of continental, non-mountain sites limits the networks ability to evaluate the $H_2$ soil sink in more detail.

We modified lines 9-12 as: ' Time series comparison of EMAC and observational data produces Pearson correlation coefficients ($r$) in excess of 0.9 at eight remote stations located in polar region and on high mid latitude islands. A further 23 sites yielded correlation coefficients between 0.7–0.9, predominantly located in remote marine stations across all latitudes and also in polar regions.'

We replaced lines 182-192 by: '38 of the 56 stations have marine characteristics (Table A1 and Fig. 1), i.e. are located coastally or on islands. 11 sites can be described as polar (latitude $\geq |60°|$), and 14 stations are positioned on mountains (i.e. a single site might have several characteristics). These stations usually measure remote, often marine air masses or free tropospheric concentrations. Just 8 sites Hungary (HUN), Mongolia (UUM) , CIBA (CIB, northern central Spain), Shangdianzi (SDZ, China), Wendover (UTA, Utah), Southern Great Plane (SGP, Oklahoma), Israel (WIS),

and Gobabeb (NMB, Namibia) fall in neither of those categories. Inner continental non-mountain stations are sparse, with none in South America and Australia. The two continental stations in Africa (ASK, and NMB) are located in the desert. There are also no sites in Central Asia. The three Mediterranean stations (CIB, LMP, and WIS) and the continental Chinese (SDZ) and Mongolian (UUM) sites are located in or close to areas with high air pollution (Lelieveld et al. (2002); Silver et al. (2025)). The Bukit site (BKT, Indonesia) is regularly impacted by biomass burning and peat fires (Yokelson et al. (2022)).

Across these 56 observational stations, the Pearson correlation coefficient ($r$) exceeds 0.9 for eight remote stations located either in the Arctic or Antarctic regions and on islands on high mid latitude. In such cases, the annual cycle of $H_2$ is modelled excellently in terms of magnitude, amplitude and seasonality. In contrast 9 stations produce correlation coefficients below 0.5. The three stations in the Mediterranean region CIB (-0.17), LMP (0.05) and WIS (-0.29) and the Chinese and the Mongolian sites SDZ (-0.2) and UUM (0.03) show coefficients close to zero or even negative. Negative $r$ values suggest that the EMAC model does not correctly capture the phasing of the annual $H_2$ cycle which results in pronounced phase mismatches or anti-correlation in Table B1 and Figures 2, B1, and B2. These sites are located in regions known for levels of high air pollution. The two tropical coastal stations BKT (0.14) and Natal (Brazil, NAT 0.14) do show rather low $r$ values compared to other tropical stations. They are impacted by biomass burning (i.e. NAT) and peat fire emissions (i.e. BKT). The comparison at the Hungarian station (HUN, 0.39) shows a considerably lower $r$ value compared to the other two central European stations Hohenpeissenberg (HPB, 0.56) and Ochsenkopf (OXK, 0.64). HPB and OXK are mountain stations and less susceptible to local mismatch in $H_2$ soil deposition, as can be seen from the much better match of amplitude and phase. The Christmas Island site (CHR, 0.43) is the only very remote tropical island station, which has a considerably low $r$ value. There the observational time series is relatively short with hardly any annual cycle. '

We modified lines 194-195 as: 'The remaining 16 stations produce correlation coefficients between 0.5–0.7'

We changed line 221 to: 'experience remote air masses'

We added in line 243: 'As mentioned above, the $H_2$ observational stations do not represent the inner region of continents well. Especially, the small number (UTA, SGP, UUM) of remote non-mountain, non-hyperarid sites, placed away from highly anthropogenically influenced regions limits its usage in looking in more detail on the parametrisation of the soil sink. At Wendover (UTA, Utah, arid cold climate, $r = 0.82$) and the Southern Great Planes (SGP, Oklahoma, humid subtropical climate, $r = 0.77$) shown in Figure 2, the model performs well. By realistically representing the magnitude, amplitude and seasonality, of the measurements this indicates that the soil sink parametrisation of $H_2$ is realistic in those regions. The poor results at the Mongolian site (UUM, arid cold climate, $r = 0.03$), Fig. B2, might be partly attributed to a resolution effect (see above).'

We modified line 307 as: 'remote observational stations and for sites measuring remote and free tropospheric air'

Furthermore, Tables A1 and B1 are supplemented with site characterisation details.

**References**

Lelieveld, J., Berresheim, H., Borrmann, S., Crutzen, P. J., Dentener, F. J., Fischer, H., Feichter, J., Flatau, P. J., Heland, J., Holzinger, R., Korrmann, R., Lawrence, M. G., Levin, Z., Markowicz, K. M., Mihalopoulos, N., Minikin, A., Ramanathan, V., de Reus, M., Roelofs, G. J., Scheeren, H. A., Sciare, J., Schlager, H., Schultz, M., Siegmund, P., Steil, B., Stephanou, E. G., Stier, P., Traub, M., Warneke, C., Williams, J., and Ziereis, H.: Global Air Pollution Crossroads over the Mediterranean, Science, 298, 794–799, https://doi.org/10.1126/science.1075457, 2002.

Paulot, F., Paynter, D., Naik, V., Malyshev, S., Menzel, R., and Horowitz, L. W.: Global modeling of hydrogen using GFDL-AM4.1: Sensitivity of soil removal and radiative forcing, International Journal of Hydrogen Energy, 46, 13 446–13 460, https://doi.org/10.1016/j.ijhydene.2021.01.088, 2021.

Silver, B., Reddington, C. L., Chen, Y., and Arnold, S. R.: A decade of China's air quality monitoring data suggests health impacts are no longer declining, Environment International, 197, 109 318, https://doi.org/https://doi.org/10.1016/j.envint.2025.109318, 2025.

Yokelson, R. J., Saharjo, B. H., Stockwell, C. E., Putra, E. I., Jayarathne, T., Akbar, A., Albar, I., Blake, D. R., Graham, L. L. B., Kurniawan, A., Meinardi, S., Ningrum, D., Nurhayati, A. D., Saad, A., Sakuntaladewi, N., Setianto, E., Simpson, I. J., Stone, E. A., Sutikno, S., Thomas, A., Ryan, K. C., and Cochrane, M. A.: Tropical peat fire emissions: 2019 field measurements in Sumatra and Borneo and synthesis with previous studies, Atmospheric Chemistry and Physics, 22, 10 173–10 194, https://doi.org/10.5194/acp-22-10173-2022, 2022.